# Mutualism increases diversity, stability, and function of multiplex networks that integrate pollinators into food webs

Kayla R. S. Hale [1✉], Fernanda S. Valdovinos [1,2] & Neo D. Martinez [3,4]

Ecosystems are composed of complex networks of many species interacting in different ways. While ecologists have long studied food webs of feeding interactions, recent studies increasingly focus on mutualistic networks including plants that exchange food for reproductive services provided by animals such as pollinators. Here, we synthesize both types of consumer-resource interactions to better understand the controversial effects of mutualism on ecosystems at the species, guild, and whole-community levels. We find that consumer-resource mechanisms underlying plant-pollinator mutualisms can increase persistence, productivity, abundance, and temporal stability of both mutualists and non-mutualists in food webs. These effects strongly increase with floral reward productivity and are qualitatively robust to variation in the prevalence of mutualism and pollinators feeding upon resources in addition to rewards. This work advances the ability of mechanistic network theory to synthesize different types of interactions and illustrates how mutualism can enhance the diversity, stability, and function of complex ecosystems.

[1] Department of Ecology and Evolutionary Biology, University of Michigan, 1105 North University Ave, Biological Sciences Building, Ann Arbor, MI 48109, USA. [2] Center for the Study of Complex Systems, University of Michigan, Weiser Hall Suite 700, 500 Church St, Ann Arbor, MI 48109, USA. [3] School of Informatics, Computing, and Engineering, Indiana University, Room 302, 919 E. 10th Street, Bloomington, IN 47408, USA. [4] Pacific Ecoinformatics and Computational Ecology Lab, Berkeley, CA 94703, USA. ✉email: kaylasal@umich.edu

As elegantly illustrated by Darwin's "tangled bank"[1], ecosystems are complex, composed of many different types of interactions among many different species. However, theory has classically predicted that complexity in terms of the number and strength of interspecific interactions destabilizes ecological systems[2]. Mutualistic interactions like those between plants and their pollinators are thought to be particularly destabilizing[3,4]. Robert May famously emphasized this point by calling mutualism an "orgy of mutual benefaction" (p. 95)[5] whose instability due to positive feedback loops helps explain why mutualism is infrequent and unimportant in natural systems[3]. Yet, mutualisms appear to be not only frequent but key to maintaining much of the biodiversity that drives ecosystems[6,7], especially agricultural ecosystems essential to human wellbeing[8,9]. Here, we address such disparities between theory and observation by developing and applying consumer-resource theory of feeding and reproductive mechanisms that integrates food webs and mutualistic networks into "multiplex" networks containing different types of interactions. We use our multiplex model to study how mutualism affects the diversity, stability, and function of complex ecosystems.

The integration pursued here benefits from long but largely separate traditions of research on feeding and mutualistic interactions[8,10]. For example, "mutualistic" feeding interactions between species with positive effects on each other (+,+), e.g. pollinators foraging on the nectar of flowering plants, are often excluded from food web data (e.g., ref. [11]), while "antagonistic" feeding interactions (+, −), e.g. herbivory and predation, are typically excluded from mutualistic networks[12]. Additionally, food web research has focused more on aquatic systems[13] where feeding interactions are strongly structured by body mass or gape size[13,14], while mutualism research has focused more on terrestrial systems where feeding interactions may be more strongly structured by other organismal traits like chemical defense and shape of mouth parts[13–15]. Within aquatic ecosystems, the allometric trophic network (ATN) theory[16,17] of food webs has leveraged body-size considerations to successfully simulate the seasonal dynamics of many interacting species[18,19] and predict the quantitative effects of experimental species manipulations[20–22]. However, the failure of these predictions in the presence of facilitation (+, 0), e.g. habitat provisioning for mussels by barnacles, highlights the need for food-web theory to better address interactions with positive effects beyond nutrition[20]. Mutualistic network theory has focused on animal-mediated pollination, an interaction involving trophic and reproductive effects[12,23], motivated in part by large agricultural and evolutionary significance[9,24,25]. Merging these distinct traditions requires a more unified approach that addresses several problems.

One significant problem is that classifying interactions based on positive and/or negative effects ignores logical inconsistencies such as when "antagonistic" herbivory or predation respectively increase plant fitness[26] or prey abundance[27] and when "mutualistic" pollinators parasitize plants by robbing their floral rewards without transferring pollen[24]. We resolve such conflicts by modeling mechanisms by which organisms interact and allowing effects to emerge from the interactions rather than asserting such effects a priori. We do this by developing consumer-resource theory that has long been applied to food web theory[16,28] and more recently applied to mutualistic interactions[29,30] with success at predicting pollinators' foraging preferences in the field[31]. Both feeding and mutualistic interactions typically involve food consumption while mutualistic interactions also often involve reproductive services provided by animals such as pollinators and seed dispersers[24,29,30]. Positive, negative, and neutral interspecific effects dynamically emerge from the benefits and costs of participating in these interactions, which often both involve the same

organisms and jointly determine much of the stability and function of ecosystems[8,10,32–36]. For example, feeding interactions such as parasitism and predation on pollinators, herbivory on animal-pollinated plants, and feeding by pollinators on animals and plant vegetation in addition to floral rewards such as nectar and pollen profoundly affect pollination dynamics, crop yields[37], and long-term sustainability of agroecosystems[38].

Another outstanding problem with understanding the joint effects of feeding and mutualistic interactions concerns contradictory conclusions of previous theoretical work. Classic "effects-based" theory has long held that mutualistic interactions are generally destabilizing (Table 1) especially at high complexity[3,4]. However, more recent theory finds that mutualisms stabilize ecological systems under conditions such as high levels of complexity of mutualism relative to antagonism in "merged" plant−pollinator and plant−herbivore networks[35,39], low levels of complexity in hierarchical networks of all types of interactions[40], or intermediate levels of mutualism when mutualistic links are randomly assigned and animals allocate effort to feeding and mutualistic interactions separately[41–43]. Contradictions among these findings may arise from different definitions of stability (i.e. local stability or persistence, Table 1) and network architectures[4,41] that often misrepresent empirically observed structure[40]. A broader problem is that narrowly focusing on stability develops inefficient theory[44] that ignores how mutualisms alter the diversity, population dynamics, and overall functioning of complex ecosystems.

To more broadly understand the ecology of mutualistic interactions, here, we follow repeated calls to synthesize different types of interactions within networks[8,10,32–36,45] by developing and applying mechanistic consumer-resource theory to "multiplex" ecological networks (Table 1)[34,36]. Our model based on this theory integrates the structure and dynamics of feeding and reproductive mechanisms from which effects of interspecific interactions emerge including predation (+, −), mutualism (+, +), and resource and apparent competition (−, −)[46,47]. We simulate network dynamics by extending Brose et al.'s ATN theory[16] to incorporate Valdovinos et al.'s theory of the exchange of food for reproductive services between plants and their pollinators[30,31]. We simulate network structures by integrating Williams and Martinez' niche model of food webs[48] with Thébault and Fontaine's model of mutualistic networks[49]. Using this multiplex model, we investigate how the presence, prevalence, and intensity of mutualism affect multiple dimensions of ecological stability[50] by assessing diversity, persistence, dynamics, and function at the species, guild, and ecosystem levels. We find that mutualism can greatly enhance ecosystem stability and function through rewards provisioning by plants with pollinators whereas mutualistic feedbacks determine ecosystem composition.

## Results

**The multiplex model.** We synthesize the structure and dynamics of feeding and reproductive interactions by integrating food webs and mutualistic networks and extending ATN theory[16,18,20,51] to include the consumption of floral rewards produced by plants and reproductive services produced by pollinators (Fig. 1). In addition to ATN theory's logistically growing "plants without pollinators", "plants with pollinators" in our multiplex model must consume reproductive services produced by their pollinators to vegetatively grow. This involves partitioning the biomass of plants with pollinators into coupled pools of vegetation and floral rewards[30] that plants produce at an energetic cost[33,52,53]. The vegetative growth rate of plants with pollinators is a saturating function of reproductive services consumed by the plants[53] determined by the quantity (consumption rate) and quality

| Table 1 Descriptions of terms. | |
| --- | --- |
| Ecosystem stability and function | We use a range of complementary metrics in a broad sense[50] to assess stability in terms of species' ability to persist with limited popuation variability and function in terms of stocks and flows of biomass. We apply the metrics both to the whole ecosystem (all species in the network) and to individual guilds in the network at near steady-state dynamics of our simulations. |
| Biomass variability (CVs) | Coefficient of Variation (CV = standard deviation/mean) of species' or guilds' biomass evaluated over the last 1000 timesteps of the simulations when their dynamics are approximately at steady state. Increased temporal stability corresponds to decreased biomass variability. |
| Consumption rate | Total amounts of biomass extracted by consumers per unit time. |
| Diversity | Number of species within a network. Networks of initial diversity $S$ are subjected to dynamical simulations, during which species may go extinct or persist. The resulting number of species is the final diversity, or simply diversity. |
| Guilds | Guilds are groups of species with similar types of consumer−resource interactions including: plants without pollinators; plants with pollinators; floral rewards of plants with pollinators (when relevant to analyze their biomass and flows separately from vegetation, Fig. 1); herbivores: species that only eat plant vegetation in the original niche-model food web (Fig. 2a); omnivores: species that eat vegetation and animals in the original niche-model food web (Fig. 2a); carnivores: species that eat only animals; added (+) herbivores/pollinators: herbivores or herbivorous pollinators added by the RO or RP treatments that consume rewards (multiplex treatments) and/or vegetation (FW treatments; Fig. 2, yellow-green nodes); added (+) omnivores/pollinators: omnivores or omnivorous pollinators added by the RP treatment that consume rewards (multiplex treatment), animals, and/or vegetation (Fig. 2e, orange nodes). Added animals include both +herbivores/pollinators and +omnivores/pollinators. |
| Local stability | The tendency of abundances of species within a system to return to their equilibrium after a very small perturbation[4]. |
| Persistence | Fraction of species that survive to the end of simulations (=initial diversity/final diversity). |
| Productivity | Total rates of biomass increase due to plant growth and food assimilated by animals minus loss due to animals' metabolic maintenance costs and plants' costs of producing rewards. |
| Steady state dynamics | Formally, dynamics in which all species have constant abundance ($dB_i/dt = dR_i/dt = 0$ for all $i$). At the end of 5000 timesteps, our systems approximate steady-state dynamics (Fig. 3) as quantified by very small variability in total ecosystem biomass over the last 1000 timesteps of the simulations (CVs < 0.0001). |
| Multiplex networks | Ecological networks that include more than one type of species interaction. Here, we focus on multiplex networks that combine food webs (including carnivorous and herbivorous feeding interactions, Fig. 2a) and pollination networks (including feeding interactions and reproductive services, Fig. 2b). |
| Rewards Only (RO) treatment | Network construction treatment in which pollinators can access floral rewards of plants w/ pollinators as their only resource (Fig. 2d). These networks are subjected to multiplex dynamics, which include pollination in addition to traditional food web dynamics. |
| Rewards Plus (RP) treatment | Network construction treatment in which pollinators can access floral rewards plus plant vegetation and/or animal biomass resources (Fig. 2e). These networks are subjected to multiplex dynamics, which include pollination in addition to traditional food web dynamics. |
| Pollination link *or* mutualistic interaction | A pollination link or mutualistic interaction between pollinator $i$ and plant w/ pollinator $j$ describes both the consumption of $j$'s floral rewards by $i$ and the reproductive services provisioning to the vegetative growth rate of $j$ by $i$ (Fig. 1). In the FW treatments, pollination links are switched to links in which $i$ consumes the vegetative biomass of $j$, i.e. to herbivory links. |
| Food Web (FW) treatments | Ecological networks with the structure similar to multiplex networks, where all pollination links are switched to herbivory links (also corresponding to zero rewards productivity). In the Rewards Only Food Web (RO FW) construction, animal $i$ is strictly an herbivore (Fig. 2d), while in the Rewards Plus Food Web (RP FW) construction, animal $i$ could be an omnivore or herbivore (Fig. 2e). These networks are subjected to traditional food web dynamics. |
| Rewards productivity ($\beta$) | Parameter in our multiplex model specifying the rate of rewards biomass produced by plants w/ pollinators per unit of their vegetative biomass. "Low" ($\beta = 0.2$) and "High" ($\beta = 1.0$) are arbitrary values chosen to illustrate two behaviors of the multiplex model compared to traditional food web dynamics. "None" corresponds to Food Web treatments. We interpret rewards productivity as a proxy for the intensity of pollination interactions. See Supplementary Fig. 2 for persistence results across a range of $\beta$ values. |
| Feedback control | To test whether transient mutualistic feedbacks isolated from rewards availability lead to the differences between our multiplex and Food Web treatments, we initialized simulations forced with rewards availability from multiplex simulations but with feedbacks (dashed and purple arrows in Fig. 1) turned off. We then observed potential changes in steady-state ecosystem stability and function. |

(fidelity) of pollinator visits[30] limited by community-wide carrying capacity[18,51]. The foraging rates and metabolic maintenance costs of all animals including pollinators scale allometrically with body size[16].

We integrated realistic food-web and mutualistic-network structures into multiplex networks by generating 102 niche-model food webs[48] of $S_f = 50$ species including exactly 20 plant species (Fig. 2a, Methods) and 238 plant−pollinator networks[49] (Fig. 2b) of varying species diversity ($S_p = A + P = 9$, 12,…,57)

and empirically observed pollinator-to-plant ratio ($A/P = 2$) and ranges of connectance and nestedness (Supplementary Methods, Supplementary Fig. 1). We pair each food web with each plant−pollinator network ($N = 102 \times 238 = 24,276$ pairs) and add the pollinators of each plant within the plant−pollinator network to its paired food web by linking the pollinators to a randomly selected plant species in the food web (Fig. 2c). Our Rewards Only (RO) treatment links only floral rewards to the pollinators (Fig. 2d). Our Rewards Plus (RP) treatment links floral rewards,

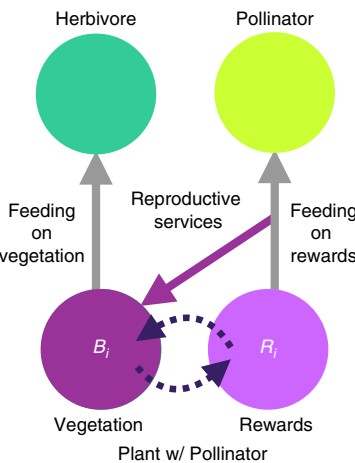

**Fig. 1 Interaction mechanisms in the multiplex model.** Interspecific and intraspecific mechanisms of feeding and reproduction combine to describe pollination mutualisms and traditional trophic interactions. Biomass of plants with pollinators is partitioned into two pools, vegetation (purple node) and floral rewards (light purple node), coupled by intraspecific dynamic feedbacks (dashed arrows). Rewards production is proportional to vegetative biomass but subject to self-limitation such that reward productivity per unit biomass decreases with increasing rewards abundance. Producing rewards incurs costs (reduced vegetative productivity), which creates tradeoffs between producing rewards to attract pollinators and benefiting from the quantity (number of visits measured as feeding rate on rewards) and quality (conspecific feeding/total feeding) of pollinators' reproductive services (purple arrow) that are required for vegetative production. At saturation, reproductive services allow plants with pollinators to potentially achieve a 25% higher per-biomass growth rate than that of plants without pollinators whose intrinsic growth rate is independent of consumers' behavior. All plants are also subjected to competition from the plant community (not shown), which reduces per-biomass vegetative growth rate close to carrying capacity. Gray arrows show herbivores feeding on vegetation and pollinators feeding on rewards.

vegetative biomass, and prey (Fig. 2e) to the pollinators. Pollinators are preyed upon by predators of herbivores in RO networks or predators of herbivores and low trophic-level omnivores in RP networks (Fig. 2d, e). As such, RO and RP treatments generate two different topological classes of multiplex networks for which we generate two groups of topologically comparable food webs (RO FW and RP FW), described below.

Sensitivity and uncertainty analyses (Supplementary Methods) revealed a pivotal role of floral rewards in determining ecological effects of mutualism (Supplementary Tables 1−3, Supplementary Fig. 2, Supplementary Discussion). We illustrate this role by presenting results from networks with High ($\beta = 1.0$), Low ($\beta = 0.2$), and no rewards productivity (Eq. 4). High and Low productivities apply to both RO and RP multiplex networks (High RO, Low RO, High RP, Low RP) containing both feeding and reproductive interactions. Treatments with no rewards productivity convert all feeding on rewards to feeding on the same plants' vegetative biomass and eliminate reproductive interactions resulting in networks with only traditional Food Web (FW) interactions where comparable plants with pollinators in multiplex networks instead grow as plants without pollinators and their former pollinators graze only on their plant partners' vegetation instead of floral rewards. This conversion transforms the added pollinators in multiplex networks into "added animals": herbivores in RO FW networks or herbivores and omnivores in RP FW networks (Fig. 2d, e). These two Food Web treatments (RO FW, RP FW) control for network structure, including the

varying numbers and trophic levels of species and links in RO and RP networks, to help elucidate the effects of mutualism in multiplex networks.

By integrating food webs of fixed size with plant−pollinator networks of varying size, the initial diversity ($S = 50 +$ added animals $= 56, 58,…, 88$) of our 24,276 networks corresponds to prevalence of herbivory in our FW treatments or prevalence of mutualism in our multiplex treatments via the fraction of plants that are animal-pollinated and the fraction and number of species and links directly involved in mutualistic interactions. This allows analysis of our results in terms of increasing initial diversity ($S$) for all treatments and increasing prevalence of mutualism for multiplex treatments. The intensity of mutualism in our treatments varies with rewards productivity (Table 1) from high (High RO, High RP) to low (Low RO, Low RP) to none (RO FW, RP FW). We simulate the 24,276 networks subjected to these six treatments by initializing each species and pool of rewards with a biomass of 10 and recording results after 5000 timesteps when species have persisted or gone extinct and system dynamics are at, or close to, steady state (Table 1, Fig. 3). We use these results to compare multiple measures of stability and function among treatments averaged over all initial diversity classes (Fig. 4) or within initial diversity classes (Figs. 5, 6) at the species, guild, and ecosystem levels (Table 1). Below, we describe the effects of the presence, prevalence, and intensity of mutualism first on biodiversity, then on ecosystem function, and finally on temporal stability.

**Diversity.** Multiplex networks with High rewards productivity (High RO, High RP) had higher average diversity (Fig. 4a) and persistence (Fig. 4b) than their counterparts with less (Low RO, Low RP) and no rewards productivity (RO FW, RP FW). Diversity and persistence were also slightly elevated in the Low RP compared to the corresponding RP FW treatment. Persistence decreased with increasing initial diversity and prevalence of mutualism in all treatments except High RO (Fig. 5c). However, these decreases were not strong enough to prevent overall increased final diversity (Fig. 5b) with increased initial diversity and mutualism (Fig. 5a) in all treatments. These results indicate that effects of mutualisms on biodiversity depend on both prevalence and intensity of mutualism expressed as rewards productivity.

Since all 20 plant species nearly always persisted in all treatments (Supplementary Fig. 3), differences in animal persistence among treatments underlie the overall patterns in diversity. Most notably, relatively high and increasing persistence of omnivores with increasing mutualism (Fig. 5e) and consistently high persistence of added pollinators (Fig. 5f, g) doubled animal diversity in High rewards treatments over that in FW and Low rewards treatments (Fig. 5b). The few carnivores (~3 initial species) made smaller contributions to elevated diversity in High rewards treatments via substantial increases in persistence with increasing mutualism (Fig. 5d). In contrast, FW and Low rewards treatments had much lower persistence of carnivores (Fig. 5d), omnivores (Fig. 5e), and added animals (Fig. 5f, g) that, except for carnivores, decreased with increasing mutualism. In the High RO treatment, the few herbivores (~5 initial species) achieved higher persistence than in the corresponding FW (Fig. 5h). However, in the other multiplex treatments (Low RO, Low RP, High RP), herbivore persistence was lower and declined dramatically with increasing mutualism.

**Function.** Similar to final diversity, the total biomass (Fig. 4c), productivity (Fig. 4d), and consumption in all multiplex treatments were comparatively higher than in FW treatments with

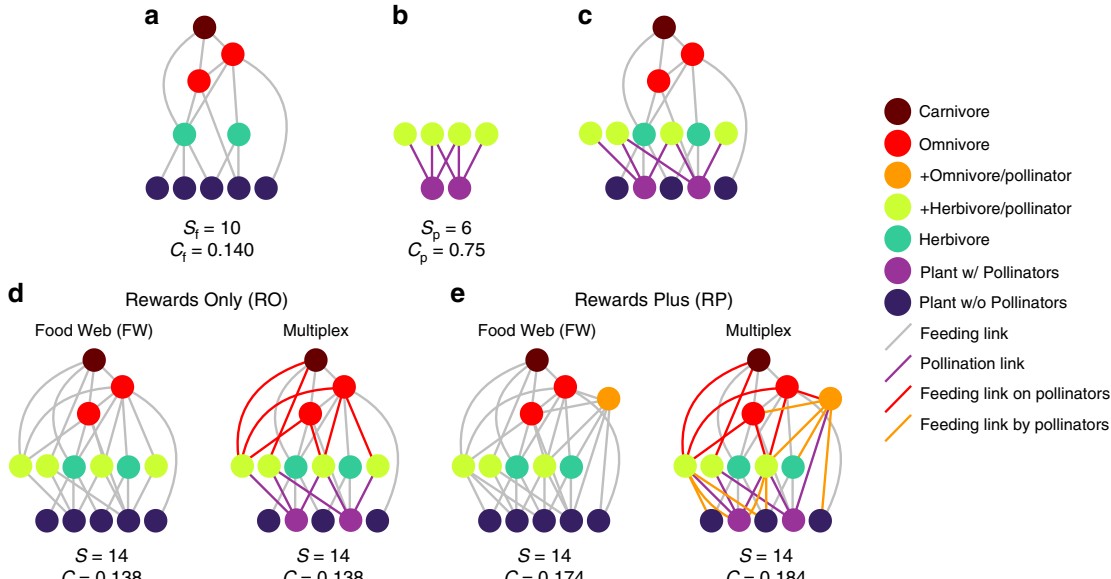

**Fig. 2 Steps for constructing food-web and multiplex-network treatments.** Nodes are vertically arranged by trophic level with plant species at the bottom and carnivores at the top. All (gray, red, orange) links represent feeding by the consumer above the resource except the bi-directional pollination links (purple, simplified from Fig. 1) that represent pollinators consuming plants' floral rewards (e.g., nectar) and plants "consuming" pollinators' reproductive services. Steps: **a** Generate food web with diversity $S_f$ and connectance $C_f$[48]. **b** Generate plant−pollinator network with diversity $S_p$ and connectance $C_p$[49]. **c** Integrate the food web and plant−pollinator network by adding pollinators and their links from (**b**) to the food web in (**a**) by randomly assigning plant species in (**a**) to become the animal-pollinated plants in (**b**). **d** Rewards Only (RO) treatments: following (**a**−**c**), construct the RO multiplex treatment by stochastically linking predators of herbivores in (**c**) to pollinators and then, for the RO FW treatment, transform pollinators into herbivores and plants with pollinators into plants without pollinators. **e** Rewards Plus (RP) treatments: following (**a**−**c**), construct the RP multiplex treatment by stochastically linking predators and diets of herbivores and herbivorous omnivores in (**c**) to pollinators and then, for the RP FW treatment, transform omnivorous and herbivorous pollinators and plants with pollinators into omnivores, herbivores, and plants without pollinators, respectively. Resulting diversity (*S*) and connectance (*C*) is shown under each network treatment (**d**, **e**). See Methods.

some exceptions in the Low RO treatment. Specifically, biomass was up to twice as high while productivity and consumption were up to an order of magnitude higher in multiplex compared to FW treatments. The Low RO treatment also had higher biomass, but lower diversity, productivity, and consumption than the corresponding RO FW treatment (Fig. 4). Overall, this indicates that both the presence and intensity of pollination mutualisms increase key measures of ecosystem function.

In all treatments, total biomass (Fig. 6b), productivity (Fig. 6c), and consumption (Fig. 6d) increased with initial diversity and prevalence of mutualism, with the strongest increases occurring in the High rewards treatments. Plant biomass decreased below carrying capacity (Fig. 6b) with increasing initial diversity in all but Low RO treatments, but this was strongly compensated for by increases in the biomass of animals and floral rewards. As might be expected, biomass and productivity of pollinators, plants with pollinators, and rewards increased with increasing mutualism in multiplex treatments. Concurrently, plant productivity increased with initial diversity and mutualism in all but Low RO treatments. Total consumption (Fig. 6d) in all treatments very closely matched total production (Fig. 6c) and was distributed similarly to that of animal species' biomass (Fig. 6b).

The higher biomass of multiplex compared to FW treatments was primarily due to increases in animal biomass (Fig. 6b), while differences in productivity were due to decreases in vegetative productivity coupled with strong increases in productivity of rewards and smaller increases in animals (Fig. 6c). These differences emerge primarily due to the interactive dynamics of rewards in which growth potential, unlike all other stocks of biomass, depends not on its own abundance but on the abundance of another component: the vegetative biomass of plants with pollinators (Methods, Eq. 6). This allows rewards to

be highly productive even when very rare whereas other network components would be unproductive or even go extinct. In FW treatments, the overall weak increases in ecosystem biomass and strong increases in total productivity with increasing initial diversity emerge from increasing herbivore biomass that reduces plant vegetation below carrying capacity (Fig. 6b), freeing plants from competition. This increases plant productivity (Fig. 6c) and animal biomass enough to lead to a net increase in total biomass with initial diversity (Fig. 6b). Multiplex treatments experience similar decreases of plant biomass and corresponding increases in vegetative plant productivity (Fig. 6b, c), but rewards productivity dramatically increases as does animal biomass. These increases are mostly due to increases in pollinator abundance that stimulate rewards productivity by depleting rewards below their self-limitation threshold. Then, animal biomass and productivity are further elevated by increases in the biomass of omnivores and carnivores that feed on the increasingly abundant pollinators. These patterns in production, consumption, and increased animal biomass are greatly enhanced in High rewards treatments.

**Stability**. We evaluated the temporal stability of our networks by analyzing coefficients of variation (CV = standard deviation/mean) of biomass during our simulations' final 1000 timesteps for each species, for the sum of each species within each guild, and for the sum of each species within the ecosystem. We calculated a species-level CV for each guild by summing each species' CV within the guild and dividing by the number of species in the guild (Fig. 6e) and a species-level CV for each ecosystem by doing the same thing for all species within the network (Fig. 4e). We calculated a guild-level CV for each guild as the CV for the total biomass in each guild (Fig. 6f), and a guild-level CV for each ecosystem by summing all guilds' CVs and dividing by the

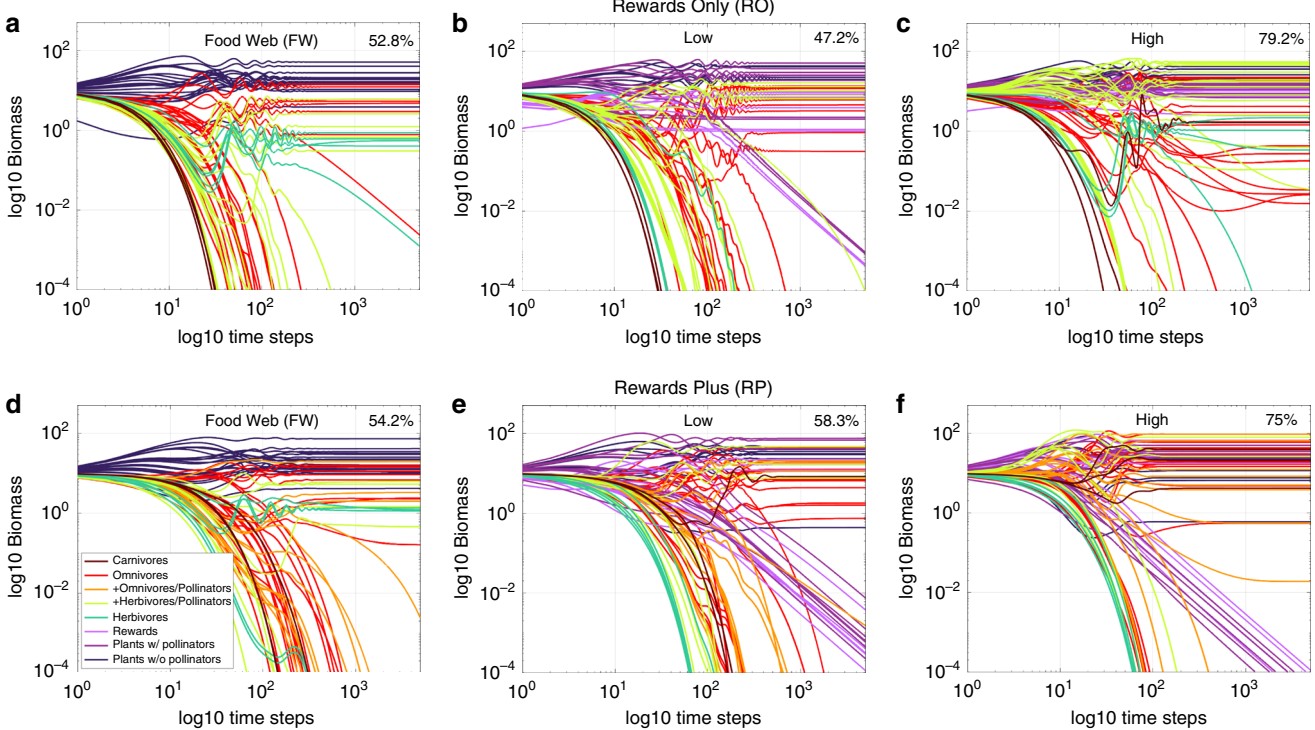

**Fig. 3 Timeseries of a 72-species ecological network subjected to six treatments.** This example describes a 50-species niche-model food web integrated with a 33-species plant−pollinator network according to the Rewards Only (RO, **a**−**c**) and Rewards Plus (RP, **d**−**f**) network treatments subjected to traditional Food Web (FW) dynamics (**a**, **d**) or multiplex dynamics with Low (**b**, **e**) or High (**c**, **f**) rewards productivity. Note that 11 plant species in the food web are chosen to represent the 11 plant species in the pollination network. Simulations last 5000 timesteps and are presented on log−log axes. Each line (colored by guild following Fig. 2) is the trajectory of a species' biomass over time. Species that fall below a biomass of $10^{-4}$ continue to extinction ($10^{-6}$). Resulting persistence is labeled in the upper right corner of each panel. Plants with pollinators are considered extinct when their vegetative biomass (purple) drops below $10^{-6}$; their rewards biomass (light purple) is strongly coupled and declines accordingly. The vast majority of species' biomasses achieve steady-state by 2000 timesteps with nearly all animal extinctions occurring before then, while several low-abundance plants with pollinators continue slow declines well past 2000 timesteps.

number of guilds (standardized across treatments, Fig. 4f). Finally, we calculated an ecosystem-level CV as the CV of the total biomass of the ecosystem.

At the ecosystem level, all treatments were exceedingly stable (CV < 0.001). In contrast, species on average were much more variable (Fig. 4e), especially in Low rewards treatments where plants with pollinators and their rewards contributed large amounts of variability (Fig. 6e). Large variability in plant and reward biomass in multiplex treatments was caused by the very low biomass of a few plant with pollinators species whose biomass decreased throughout the end of the simulations (e.g. Fig. 3b, e, f). This low and decreasing biomass yields large CVs at the species level but contributes very little to guild-level variation (Fig. 6e, f) due to the tiny fraction of their guild's biomass comprised by these very rare species. In FW treatments, where extinctions occur relatively early (Supplementary Figs. 4, 5), species-level and guild-level variation are comparable (Fig. 4e, f). In contrast, average guild-level variation (Fig. 4f) is only a fraction of the species-level variation (Fig. 4e) in multiplex treatments. Variation at both the species and guild levels decreases with initial diversity and mutualism in all treatments except for species-level variation in High RP networks where large decreases in animal variability only partly compensate for larger increases in rewards variability (Fig. 6e, f). High RP networks are by far the most stable at the guild level however, both on average (Fig. 4f) and with increasing mutualism (Fig. 6f). Overall, mutualism broadly stabilizes the dynamics of multiplex networks by reducing variability of animal populations compared to those in FW treatments.

**Overall effects of mutualism.** Excepting Low RO networks, multiplex treatments had higher average diversity, persistence, biomass, productivity, and consumption than their FW counterparts (Fig. 4). Multiplex treatments were also more temporally stable than FW treatments at the guild level and, for animal populations, at the species and guild levels. Interestingly, though Low RO networks displayed lower average diversity, persistence, productivity, and overall species-level stability than FW treatments, these effects were ameliorated in Low RP networks, in which pollinators had additional food available to them in the form of plant and animal resources. This suggests that the positive effects of mutualism are reasonably restricted to systems that provide sufficient food for mutualistic animals to survive (Supplementary Discussion) and that the low persistence of the many added pollinators in Low RO networks (Fig. 5g) throttles mutualisms from more generally increasing ecosystem diversity, function, and stability beyond that of FW treatments.

**Mutualistic feedbacks.** We studied the degree to which mutualistic feedbacks affect ecosystems beyond broadly providing food for animals by developing "feedback controls". At steady state of the multiplex simulations, plants' production of rewards interacts with their vegetative production and their pollinators' consumption. These interactions emerge from the dynamic feedbacks between plants and pollinators whereby plants produce rewards, which pollinators consume while providing reproductive services, which increase vegetative growth rate, which affects vegetative

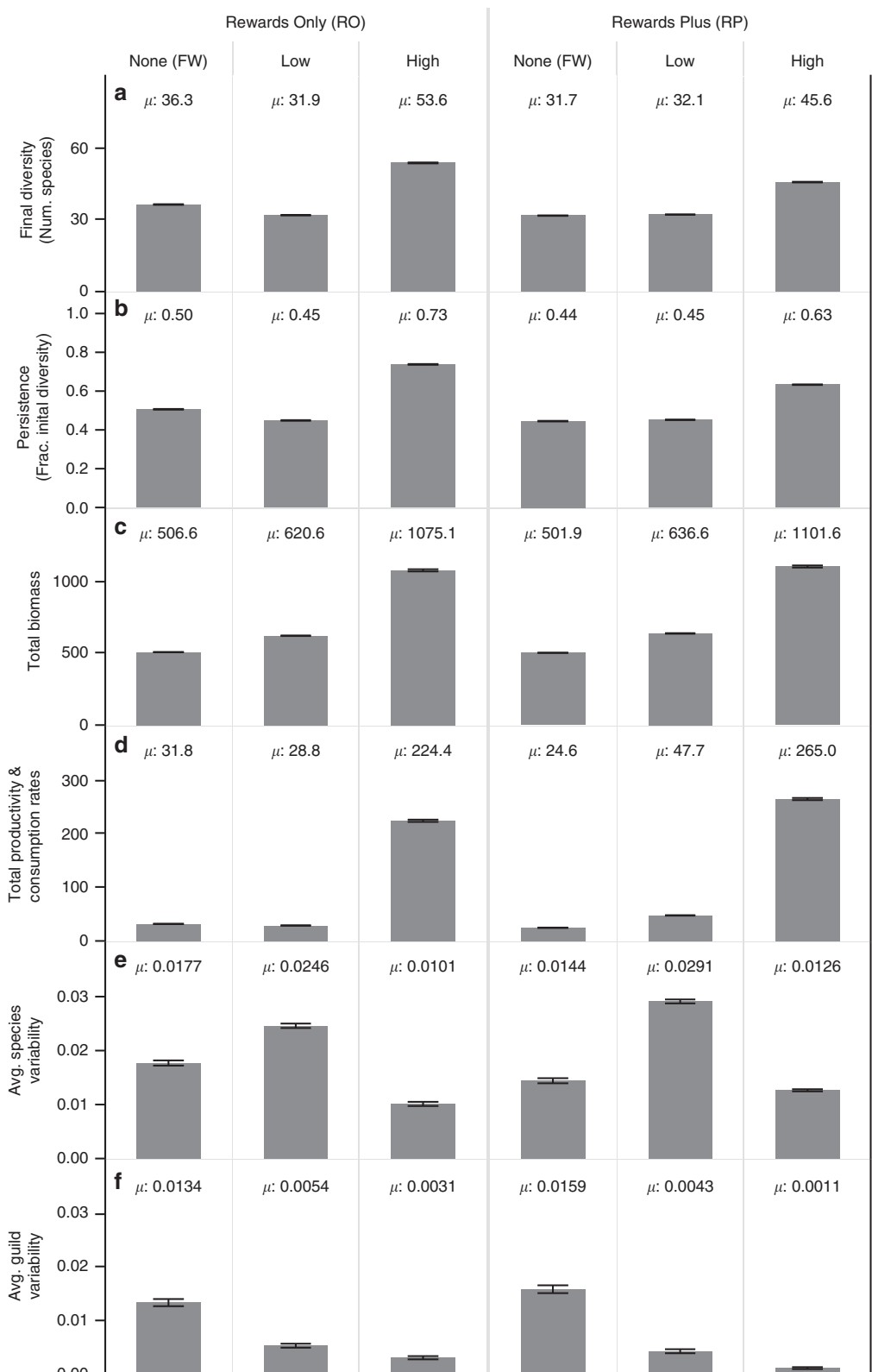

**Fig. 4 Overall effects of mutualism on stability and function in complex ecosystems.** Column headings label the treatments described in Fig. 3. Mutualism is absent in Food Web (FW) treatments and present in corresponding multiplex treatments. Mutualisms are less intense in Low than in High rewards productivity treatments. Gray bars and associated $\mu$'s describe means over all levels of initial diversity for all networks or "ecosystems" within each treatment at the end of $N = 24{,}276$ simulations. Shown are the total **a** diversity, **b** persistence, **c** biomass, **d** productivity and consumption rates, and the mean CVs of biomass of all **e** species within each ecosystem and **f** guilds within each ecosystem averaged over all the ecosystems within each treatment. Black error bars are 95% confidence intervals. Total rates of productivity and consumption were approximately equal (i.e. at steady state, all production is being consumed) so they are shown in one row (**d**).

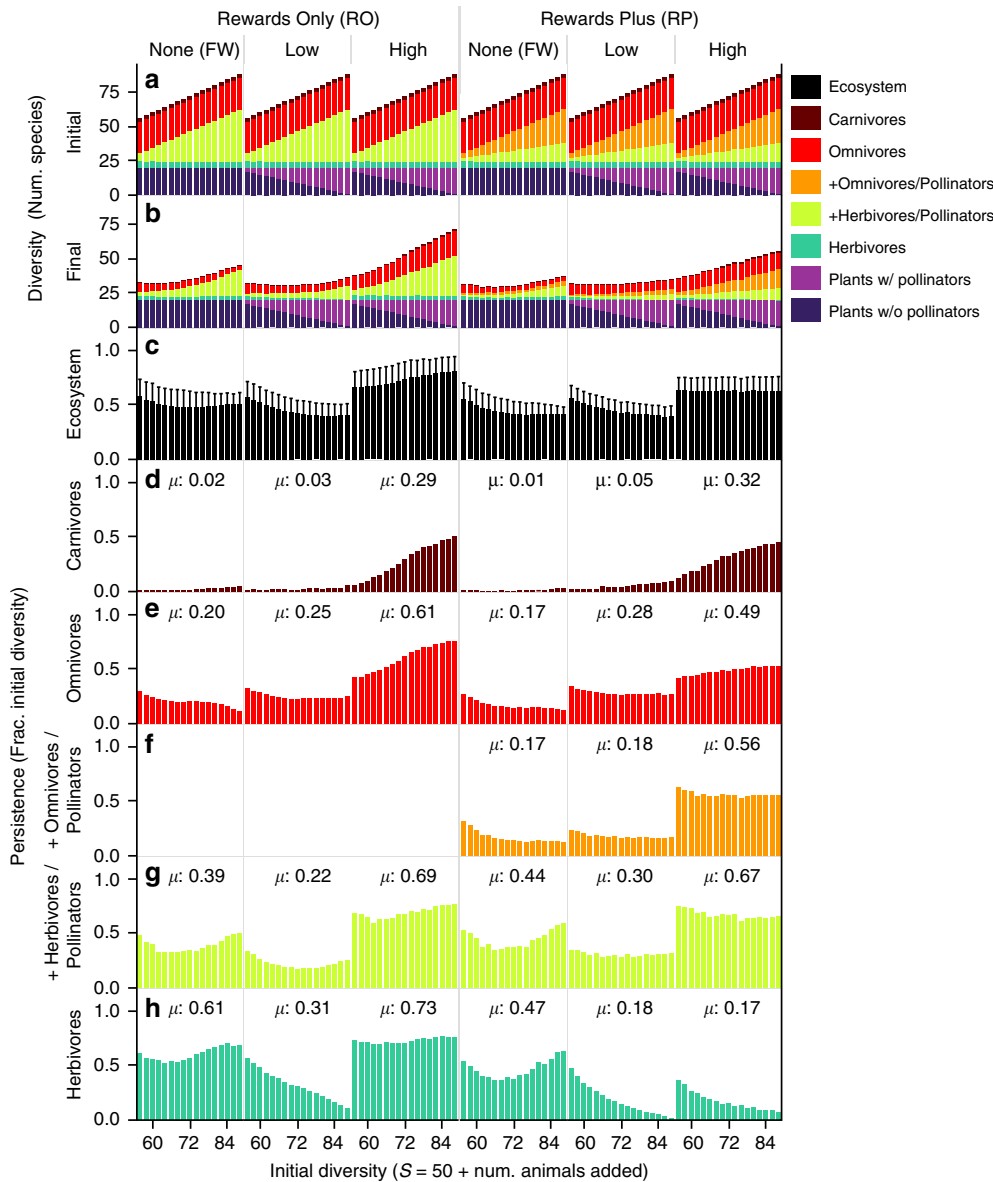

**Fig. 5 Increasing mutualism affects diversity and persistence.** Colors represent guilds of species described in Fig. 2 and Table 1. Initial diversity (*S*) on the *x*-axis and in (**a**) describes the initial number of species in food webs plus added herbivores and omnivores in the Food Web (FW) treatments or pollinators in the multiplex treatments. Increasing *S* corresponds both to an increasing number of added (+) animals and to an increasing fraction of the 20 plants without pollinators that are assigned to be plants with pollinators, and thus to increasing prevalence of mutualism in multiplex treatments. Bars show mean values for networks of a given *S* in increments of two species. Stacked bar graphs show the mean number of species in each guild (colors) that contribute to **a** initial or **b** final ecosystem diversity. Persistence, the fraction of the initial diversity that persists to the end of the simulations, is shown for **c** the entire ecosystem (i.e. network of species) and **d**−**h** for each guild of animals. **c** Error bars show standard deviations. **d**−**h** $\mu$ is the mean guild persistence over all $N = 24{,}276$ simulations in each treatment. Plants nearly always persist in our treatments (Supplementary Fig. 3), so their persistence is not shown.

biomass, which affects rewards productivity, etc. (Fig. 1). Our feedback controls are nonmutualistic systems initialized with rewards and forced to produce rewards at rates seen in a steady-state mutualistic system, but without the mutualistic feedbacks (dashed and purple arrows in Fig. 1 removed). This allows us to test whether the additional biomass produced by plants with pollinators is the sole cause of diversity, stability, and function in our multiplex networks or whether plant−pollinator feedbacks are required for these effects (Methods).

The overall ecosystem diversity, persistence, biomass, and productivity in our feedback controls equilibrate to similar values as in the multiplex simulations (Supplementary Figs. 6, 7). However, ecosystem composition in controls differed from that of

multiplex treatments. Eliminating dynamic feedbacks increased vegetative biomass of plants with pollinators, decreased biomass of plants without pollinators, and decreased persistence and biomass of omnivores and herbivores (Supplementary Fig. 8). These guild-level differences were tiny in the High RP treatment but much larger in the RO and Low RP treatments. This pattern suggests that the combination of sufficient rewards productivity and increased trophic connectedness of mutualists in High RP networks dampen the effects of mutualistic feedbacks.

Overall, our results suggest that the added productivity of mutualistic rewards drive our observations of ecosystem stability and function in the multiplex treatments. However, our results also suggest that the dynamics of mutualistic feedbacks alter the

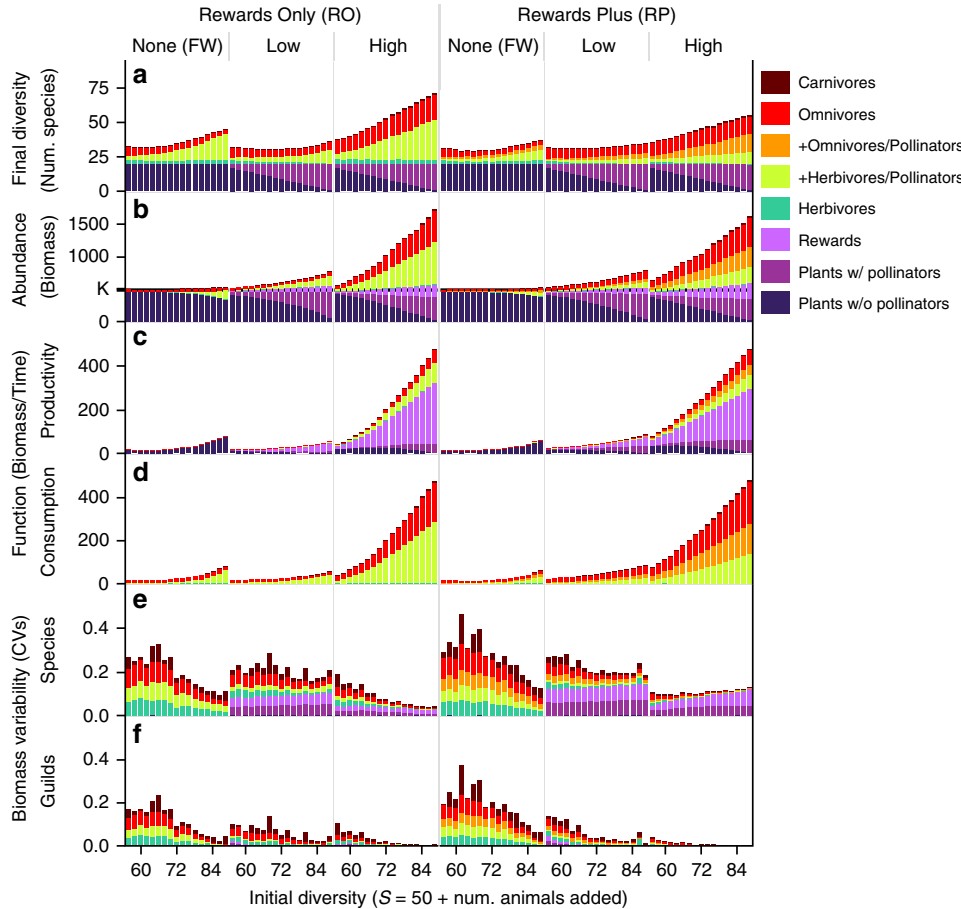

**Fig. 6 Increasing mutualism affects abundance, function, and temporal stability.** Formatting follows Fig. 5 with the addition of floral rewards shown in lightest purple. **a** Final diversity from Fig. 5 is shown again for reference. Stacked bar graphs show the contribution of each guild (colors) to total ecosystem **b** abundance, **c** productivity, **d** consumption, and average variability over time of **e** species and **f** guilds, with colors representing the average variability of the guild in (**f**) or species within the guild in (**e**). **b** The community-wide carrying capacity for plant vegetative biomass ($K$) is marked on the y-axis; total plant vegetative biomass (summed vegetation of plants with and without pollinators) does not exceed this value.

composition of ecosystems by increasing biomass and persistence of consumers, particularly omnivores, and decreasing biomass of plants with pollinators (Supplementary Discussion).

## Discussion

Our investigation of the effects of mutualism on the diversity, stability, and ecosystem function of realistically complex ecological networks used six treatments that varied the intensity (no, low, and high productivity of floral rewards) and prevalence (number and fraction of species directly involved in mutualisms) of mutualistic interactions between plants and their pollinators that either ate only floral rewards (rewards only, RO) or other sources of biomass in addition to floral rewards (rewards plus, RP). We found that adding more intense mutualisms strongly increases the diversity, persistence, productivity, consumption, biomass, network complexity (number of species and interactions), and temporal stability of biomass over that in networks with weaker or no mutualisms (Fig. 4). With a few exceptions, these effects increased with the prevalence of both strong and weak mutualisms (Figs. 5, 6). Perhaps most strikingly, overall persistence increased with the prevalence of strong mutualisms involving RO pollinators (Fig. 5c, High RO). Temporal stability more generally increased in all six treatments with diversity and prevalence of mutualism at the levels of species (Fig. 6e) and guilds (Fig. 6f) except for the slightly negative relationship at the species-level in networks with strong mutualisms

involving RP pollinators (Fig. 6e, High RP). With one exception (Low RO productivity), networks with the most prevalent mutualistic interactions had higher abundance (Fig. 6b), productivity (Fig. 6c), and guild-level stability (Fig. 6d) than in corresponding networks with no mutualism.

These increases of stability and function due to increases in mutualism are broadly consistent with empirical observations of the effects of increased pollinator diversity in blueberry, watermelon, and other agroecosystems[54–57]. Though consistent with empirical observations, our results are unexpected in light of classic theory that mutualism decreases stability to small perturbations around assumed equilibria[2–4], which we did not assess. Instead, we more comprehensively addressed both stability in terms of diversity, persistence, and biomass variability and also function in terms of productivity, consumption, and biomass abundance at species, guild, and ecosystem levels of organization (Table 1)[50]. Mutualism tends to increase stability and ecosystem function according to all of our measures in treatments with stronger mutualistic interactions and by several metrics in treatments with weaker mutualisms (Fig. 4).

Examining our results in more detail indicates that these strong positive effects of mutualism on ecosystems emerge because mutualistic rewards at Low and even more so at High productivity levels stabilize ecological networks by increasing persistence of omnivores (Fig. 5e), omnivorous pollinators (Fig. 5f), and carnivores (Fig. 5d) above that of comparable species in food

webs of only feeding interactions (FW treatments). In contrast, pollinators that only consume rewards (RO treatments) exceed the persistence of comparable herbivores in FW treatments at High rewards productivity only. Our feedback controls show that these positive effects of mutualism are not fully reproduced by traditional food web dynamics when supplementing food webs with rewards productivity similar to that in multiplex networks. The removal of dynamic feedbacks between pollinators and animal-pollinated plants alters species composition by reducing persistence and abundance of omnivores but increasing abundance of plants formerly dynamically partnered with pollinators (Supplementary Figs. 6−9). Thus, the combination of dynamic feedbacks and mutualistic interaction strength, in terms of rewards productivity, leads to the increased stability and function we observed in our multiplex ecological networks.

Our theory embraces Levin's assertion that "The key to prediction and understanding lies in the elucidation of mechanisms underlying observed patterns" (p. 1943)[58] by using trophic and reproductive mechanisms underlying the observed structure and dynamics of multiplex networks to understand and predict how mutalism affects complex ecosystems. Previous theory obscures key dynamics of such mechanisms by assuming their effects[23,47,59]. For example, the few studies of ecological networks involving both feeding and mutualistic interactions assert species have constant per-capita direct effects on each other's fitness or abundance[36,45]. Such effects can rarely be reproducibly measured due to, e.g., context-dependence and temporal variability[59,60]. We instead focused on mechanisms such as more-reproducible rates of production and consumption of food in terms of biomass[18,61] and reproductive services in terms of pollinator visits[36]. We also focused on more realistic (i.e., empirically corroborated) network architectures—as have others with results similar to ours. For example, García-Callejas et al.[40] found that mutualism increases or leaves unaffected persistence of ecological networks containing different types of interactions structured according to realistic species abundance distributions among different trophic levels. Several studies of near-equilibrium stability similarly find that destabilizing effects of mutualism may be overstated in models of fewer species[32] and interaction types[41] compared to more realistic ecological networks. However, unlike studies[41,42] that require the "peculiar constraint" (p. 528)[43] that animals' interaction efforts are allocated separately between mutualistic and nonmutualistic interactions, we find that mutualism is stabilizing according to empirically relevant metrics (Table 1) when unpartitioned effort is allocated to both types of interactions.

Our multiplex treatments may bound the diet breadth of pollinators in the field by providing pollinators unrealistically few resources in RO networks and unrealistically many resources in RP networks. This suggests results intermediate between RO and RP networks may best predict empirical observations. For example, in both RO and RP treatments, High rewards productivity increases the persistence, biomass, and productivity on average of all consumer guilds over that in Low rewards networks (Figs. 5d−h, 6b, c) excepting the decrease in herbivore persistence in RP networks (Fig. 5h). This suggests that, in natural systems, we may expect the weakest increases in persistence, biomass, and productivity due to increased prevalence of mutualism to occur among herbivores. The different effects of our treatments on herbivores may be due to RP pollinators' additional herbivorous and carnivorous feeding links compared to RO pollinators. These additional links increase competition and predation pressure on herbivores by pollinators, omnivores, and carnivores. More broadly, our results suggest that, as the prevalence of pollination mutualisms increase, the diversity and biomass of upper trophic-level consumers will increase while our predictions for the effects on herbivores are less certain.

Key limitations of our work concern how the network architecture and bioenergetic parameters in our models match those seen in nature. While the multitrophic and mutualistic components of our multiplex network structures correspond to empirical patterns, we lack observations of their interconnection into multiplex networks with which to test against our simulated networks[8,45], though recent multiplex networks for rocky intertidal systems that include facilitation suggest progress in this direction[36]. Regarding dynamics, the metabolic rates of animals should be better related to the intrinsic productivities of plants. Most such estimates come from aquatic ecosystems that often differ from those in aboveground terrestrial systems[13,14]. The cryptic yet critically important rates determining reward availability are also only rarely estimated. For example, Baude et al.[62] suggest that nectar productivity of meadows in the UK can be up to ~5−10% of net primary productivity and Adgaba et al.[63] estimate nearly 1000 kg ha$^{-1}$ of floral sugar is produced in a 5-month flowering season by a single tree species. Pollen production may significantly increase such estimates and accounting for seasonality of reward production and pollinator activity could greatly decrease estimates of resource demand needed to sustain pollinators. This highlights the need for improved theory of seasonal effects on both food webs[18] and mutualistic networks[64]. Bioenergetic costs of reward production compared to vegetation production should also be considered. Accounting for such anabolic efficiencies greatly increased the ability of ATN theory[16,18,20] to predict realistic stocks and flows of carbon and energy in complex food webs[18,65]. Compared to the relatively complex compounds that comprise vegetative biomass, efficiencies of synthesizing simple sugars that provide nearly all the usable energy in nectar may be much higher. Such efficiencies are suggested by estimates for animals that indicate, for example, the energetic efficiency of producing milk is almost six times than that of beef[66]. Attending to these limiting aspects of the structure and function of multiplex networks may advance research on networks including plants and pollinators as well as those involving coral, mycorrhizal fungi, and frugivores that disperse seeds, all of which involve the exchange of autotrophic food for increased growth of primary producers. More broadly, our theoretical development shows how nontrophic effects can more generally include effects that directly depend on interaction dynamics, such as quality and quantity of pollinator visits, in addition to depending on the abundance of interactors as in previous work[10,40,41,67].

We have advanced theory on multiplex networks in order to explore the effects of mutualism on ecological systems. Our focus on reproductive interactions follows previous extensions of ATN theory involving plant nutrients[20], detritus[18], ontogenetic niche shifts[68], environmental stochasticity[69], fishing[51,68], economics[51] and other nontrophic effects[67] integrated into food webs comprised of feeding interactions distinguished by their relative body sizes[16] such as diseases[11], parasites[11,70], parasitoids[11], and predators[13]. Such integration of multiple interaction types into multiplex networks is increasingly recognized as an effective means of accommodating different mechanisms responsible for ecosystem structure and function[8,10,36,67]. Our application to mutualistic mechanisms finds a striking ability of mutualism to enhance ecosystems including their diversity, stability, and function when mutualistic rewards suffice to sustain mutualistic partners. Further work incorporating mutualism into multiplex network theory (e.g. mutualisms between zooxanthellae and coral polyps or plants and mycorrhizal fungi) may illuminate whether mutualisms more generally enhance other systems. Such research would help compensate for ecologists' emphasis on competition by elucidating the much less studied roles of mutualistic and other facilitative interactions[24] in biodiversity maintenance[6], ecosystem function[7], and evolution[25].

## Methods

**Network architecture.** We created multiplex networks (Fig. 2) by generating food webs using the "niche model" (Fig. 2a) parameterized with 50 species ($S_f = 50$) and 10% directed connectance ($C_f = L_f/S_f^2 = 0.1$, where $L_f$ is the number of feeding links)[48]. The niche model stochastically assigns each species $i$ three traits: (1) a niche value ($n_i$) drawn randomly from a uniform distribution between 0 and 1, (2) a feeding range ($r_i$) where $r_i = x n_i$ and $x$ is drawn from randomly from a beta distribution with expected value $2C_f$, and (3) a feeding center ($c_i$) drawn randomly from a uniform distribution between $r_i/2$ and $\min(n_i, 1 - r_i/2)$. Species $i$ feeds on $j$ if $n_j$ falls within $i$'s feeding interval $[c_i - r_i, c_i + r_i]$. We selected niche-model food webs with $0.0976 < C_f < 0.1024$ that were comprised of 50 species ($S_f = 50$), of which exactly 20 were plants and five were herbivores that only feed on plants (i.e. have trophic level [TL] = 2), yielding 102 food webs. We also generated plant−pollinator networks using a stochastic model[49] with $3−19$ plant-with-pollinator species ($P$) and exactly twice as many animal−pollinator species ($A = 2 \times P$) to maintain pollinators' average resource availability in networks of increasing diversity. This yielded approximately 14 networks within each of 17 diversity classes ranging from 9 to 57 species ($S_p = P + A = 9, 12, …, 57$) for a total of $14 \times 17 = 238$ plant−pollinator networks that covered the empirically observed range of nestedness (Fig. 2b, see Supplementary Methods for more details). We constrained the number of pollination links ($L_p$) to ensure that pollination connectance ($C_p = L_p/PA$) broadly decreased as $S_p$ increased in an empirically realistic manner (Supplementary Fig. 1a). We integrated each of the 238 plant−pollinator networks with one of the 102 food webs yielding $N = 238 \times 102 = 24{,}276$ networks of increasing species diversity ($S = S_f + A = 56, 58, …, 88$). We did this by randomly choosing $P$ of the 20 plant species already in the food web and assigning the $A$ pollinators to those $P$ plant species as determined by the plant−pollinator network (Fig. 2c). This left $20 – P$ plant species without pollinators.

We linked pollinators to consumers in the food web in Rewards Only (RO) treatments by setting each pollinator's $n_i$ to ±5% of the $n_i$ of a randomly selected strict herbivore (TL = 2) from the food web (Fig. 2d). Pollinators' $r_i$ and $c_i$ were set to zero. This causes pollinators to be preyed upon by predators similar to predators of herbivores and to consume only floral rewards as determined by the plant−pollinator network. Because the connectance ($C_p$) of our simulated plant−pollinator networks decreases with increasing diversity ($S_p$) and pollinators have no other resources, connectance ($C = L/S^2$, where $L$ is the total number of links) decreases on average from 0.091 to 0.06 as initial species diversity ($S$) increases from 56 to 88 in the RO multiplex and corresponding Food Web (FW) treatments (Supplementary Fig. 10). In the Rewards Plus (RP) treatments (Fig. 2e), we set each pollinator's $n_i$, $r_i$ and $c_i$ to ±5% of the corresponding $n_i$, $r_i$ and $c_i$ of a randomly selected herbivore or omnivore that eats plants ($2 \le$ TL $\le 2.3$). The RP treatment links herbivorous and omnivorous pollinators to food webs (Fig. 2c), which maintains a constant average connectance ($C = 0.102$) with increasing $S$ (Supplementary Fig. 10). Feeding on both vegetation and floral rewards of the same plant species allows two links between plants and pollinators in RP networks. The corresponding FW treatment has slightly less $C$ than the RP multiplex network because the FW eliminates the link to rewards and maintains only the herbivory link (Fig. 2d, Supplementary Fig. 10). We ignore this issue to simplify comparisons between all treatments.

Overall, as initial species diversity ($S$) increases (Fig. 4a), plants with pollinators in the multiplex networks increase from 3 to 19 of the 20 total plant species and mutualistic interactions increase from directly involving $16−65\%$ of species in the networks. Correspondingly, initial herbivory in Food Web (FW) treatments increases from directly involving approximately half to three quarters of the species in the networks. We thus analyze how outputs vary with increasing initial diversity, which corresponds to increasing prevalence of mutualism in multiplex treatments or increasing herbivory in FW treatments.

**Network dynamics.** To model multiplex dynamics, we extended ATN theory[16,18,20,51] by integrating a consumer-resource approach to pollination mutualisms in which pollinators feed on floral rewards ($R$) and plants consume reproductive services produced by plants[30,31]. Plants benefit from pollinators depending upon on the quantity and quality of pollinators' visits in terms of the rate at which pollinators consume plants' rewards and the fidelity of pollinators' visits to conspecific plants[30,31]. Pollinators in RP treatments also feed on species' biomass according to ATN theory.

More specifically, ATN theory models the change in biomass $B_i$ over time $t$ for consumer $i$ as

$$\frac{dB_i}{dt} = \sum_{j \in \text{resources}} C_{ij}(B_j) - x_i B_i - \sum_{j \in \text{consumers}} C_{ji}(B_i)/e_{ji}, \qquad (1)$$

where $x_i$ is the allometrically scaled mass-specific metabolic rate of species $i$ and $e_{ji}$ is the assimilation efficiency of species $j$ eating $i$. $C_{ij}$ is the rate of species $i$ assimilating $B_j$, the biomass of species $j$:

$$C_{ij}(B_j) = x_i y_{ij} B_i F_{ij}(B_j), \qquad (2)$$

where $y_{ij}$ is the allometrically scaled maximum metabolic-specific consumption

rate. $F_{ij}(B_j)$ is the functional response for $i$ eating $j$:

$$F_{ij}(B_j) = \frac{\omega_{ij} B_j^h}{B_{0ij}^h + \sum_{k \in \text{resources}} \omega_{ik} B_k^h}, \qquad (3)$$

where $\omega_{ij}$ is $i$'s relative preference for $j$, $h$ is the Hill coefficient[71], and $B_{0ij}$ is the "half-saturation" density of resource $j$ at which $i$'s consumption rate is half $y_{ij}$[18]. The form of the preference term, $\omega_{ij}$ determines if a trophic generalist ($i$) is treated either as a "strong generalist" ($\omega_{ij} = 1$) or "weak generalist" ($\omega_{ij} = 1/(\text{num. species in } i\text{'s diet})$[72]. Here, we present results only for weak generalists that search for each of their resources equally even if one or more of their resources are extinct. Equation 3 is a Type II functional response when $h = 1$ and a Type III response when $h = 2$. We use $h = 1.5$ for a weak Type III response[71].

We use ATN theory's logistic growth model[18] to simulate biomass dynamics of plants without pollinators as:

$$\frac{dB_i}{dt} = \left(1 - \frac{1}{K} \sum_{j \in \text{plants}} B_j\right) r_i B_i - \sum_{j \in \text{consumers}} C_{ji}(B_i)/e_{ji}, \qquad (4)$$

where $r_i$ is the maximum mass-specific growth rate of plant $i$, and $K$ is the carrying capacity of the plant community. For plant with pollinators $i$ (Fig. 1), we model its vegetative biomass dynamics as:

$$\frac{dB_i}{dt} = \left(1 - \frac{1}{K} \sum_{j \in \text{plants}} B_j\right) r_i B_i P(R_i) - \sum_{j \in \text{consumers}} C_{ji}(B_i)/e_{ji} - \kappa_i(\beta_i B_i - s_i R_i) \qquad (5)$$

and the dynamics of its floral rewards biomass as:

$$\frac{dR_i}{dt} = \beta_i B_i - s_i R_i - \sum_{j \in \text{pollinators}} C_{ji}(R_i), \qquad (6)$$

where $\beta_i$ is the production rate of floral rewards, $s_i$ is the self-limitation rate of floral reward production, and $\kappa_i$ is the cost of producing rewards in terms of total vegetative growth. $P(R_i)$ is the functional response describing how benefit to $i$ accrues due to reproductive services provided by $i$'s pollinators:

$$P(R_i) = f\left( \sum_{j \in \text{pollinators}} \overbrace{C_{ji}(R_i)}^{\text{quantity}} \overbrace{\frac{C_{ji}(R_i)}{\sum_{k \in \text{resources}} C_{jk}(B_k \text{ or } R_k)}}^{\text{quality}} \right) \qquad (7)$$

which is a function of the quantity and quality of pollination provided by pollinator $j$. Quantity is $j$'s consumption rate on $i$'s floral rewards. Quality is $j$'s consumption of $i$'s rewards as compared to $j$'s consumption of all the resources it consumes. Quality is therefore $j$'s relative consumption rate of $i$'s floral rewards, a measure of $j$'s fidelity that ensures typically specialist pollinators provide higher quality services than generalist pollinators by, for example, depositing higher concentrations of conspecific pollen[30]. The form of the functional response describing benefit accrual due to animal-pollination ($f$) reflects the assertion that reproductive services saturate[53] at 1 according to: reproductive services/(0.05 + reproductive services). As $P(R_i)$ approaches 1, the realized growth rate of plant with pollinators $i$'s vegetative component approaches $r_i$, its maximum growth rate.

Pollinators follow the dynamics typical of ATN consumers (Eq. 1) with the exception that they access rewards biomass $R_i$ instead of $B_i$ in RO treatments (Eq. 8) or in addition to the biomass of other resource species (Eq. 9) in RP treatments:

$$\frac{dB_i}{dt} = \sum_{j \in \text{resources}} C_{ij}(R_j) - x_i B_i - \sum_{j \in \text{consumers}} C_{ji}(B_i)/e_{ji}, \qquad (8)$$

$$\frac{dB_i}{dt} = \sum_{j \in \text{resources}} C_{ij}(R_j \text{ and } B_j) - x_i B_i - \sum_{j \in \text{consumers}} C_{ji}(B_i)/e_{ji}. \qquad (9)$$

**Parameterization.** Vital rates for consumers follow previously described allometric scaling for invertebrates[51]. Specifically, we set plant species' "body mass" to a reference value ($m_i = 1$)[16] and calculated consumers' body mass as $m_i = Z_i^{\text{swTL}i - 1}$, where swTL$_i$ is $i$'s short-weighted trophic level[73] and $Z_i$ is $i$'s average consumer-resource body size ratio sampled from a lognormal distribution with mean = 10 and standard deviation = 100. Then, for $i$ eating $j$, $i$'s mass-specific metabolic rate ($x_i$) is $0.314 \, m_i^{-0.25}$, its maximum metabolic-specific consumption rate ($y_{ij}$) is 10, and its assimilation efficiency ($e_{ij}$) is 0.85 if $j$ is an animal or 0.66 if $j$ is plant vegetation. We set the maximum mass-specific growth rate ($r_i$) of plant $i$ to be 0.8 for plants without pollinators or 1.0 for plants with pollinators, so that when sufficient reproductive services are provisioned by pollinators, the mass-specific growth rate of plants with pollinators is comparable or can even exceed that of the plants without pollinators.

The remaining parameters are not allometrically constrained. We assigned a "half-saturation" density for consumers of species' biomass or rewards of $B_0 = 60$

or 30, respectively. This reflects decreased "handling time" for rewards compared to typically more defended vegetation. We also assigned a Hill coefficient of $h = 1.5$, a community-wide carrying capacity for plant vegetative biomass of $K = 480$, and an assimilation efficiency of $e_{ij} = 1.0$ for pollinator species $i$ consuming the floral rewards of $j$. For plants with pollinators, we used a rewards production rate of $\beta_i = 0.2$ or $1.0$ (Low or High rewards productivity treatments, respectively), a self-limitation rate of $s_i = 0.4$, and a vegetative cost of rewards production of $\kappa_i = 0.1$. In FW treatments, rewards are zeroed out ($\beta_i = 0$) and all plants are parameterized so that they behave as plants without pollinators while pollinators are parameterized as "added animals" (herbivores or omnivores) that consume vegetation with the associated lower assimilation efficiency ($e_{ij} = 0.66$) and higher half-saturation density ($B_0 = 60$) but have otherwise unchanged vital rates. See Supplementary Table 4 for a summary of model parameters and values.

**Simulations**. We simulated each of our $N = 24{,}276$ networks subjected to each six treatments (High RO, Low RO, RO FW, High RP, Low RP, RP FW) for a total of 145,656 simulations. We used MATLAB's[74] differential equation solvers (ode15s for the multiplex treatments and ode45 for FWs) to simulate these networks for 5000 timesteps (Fig. 3). By 2000 timesteps, the simulations were approximately at dynamical steady state, which we assessed through small changes in persistence with increased simulation length. Specifically, persistence decreased by 5% on average between 2000 and 500,000 timesteps in a sample of 90 networks from each treatment (Supplementary Fig. 4). We initialized all biomasses ($B_i$ and $R_i$) to 10 and used an extinction threshold of $B_i < 10^{-6}$. Statistical analyses were performed in JMP 14[75]. Our results are qualitatively robust to simulation length (Supplementary Figs. 4, 5). Sensitivity of our results to parameter variation are reported in the Supplementary Information (Supplementary Tables 1, 2) and qualitative effects of each parameter are summarized in Supplementary Table 3.

**Outputs**. We quantified ecosystem stability and function using species persistence, biomass, productivity, consumption, and variability at or near the end of the simulations, when the dynamics were approximately at steady state (Table 1). We calculated these metrics for the whole ecosystem (Fig. 4) and for seven guilds of species (Figs. 5, 6). Two guilds are self-evidently described as species of plants without pollinators and plants with pollinators. Herbivores, omnivores, and carnivores refer only to species present in the niche-model food webs prior to integrating animals from plant−pollinator networks in Fig. 2c. Herbivores eat only vegetative biomass. Omnivores eat vegetation and animals. Carnivores eat only animals. The meanings of the two remaining guilds (collectively referred to as the "added animals") depend on the treatment that adds them to the food web. Added herbivores/pollinators refer to herbivores added by the FW treatments, pollinators added by the RO or RP multiplex treatments that consume only rewards, and pollinators added by the RP multiplex treatment that consume rewards and vegetation. Added omnivores/pollinators refer to omnivores added by the RP FW treatment and pollinators added by the RP multiplex treatment that consume rewards, other animals, and potentially vegetation. When relevant (e.g. in Fig. 6), we considered the rewards biomass of all plants with pollinators as an eighth guild.

We calculated all outputs at the end of the simulations (timestep 5000) except for biomass variability, which we calculated over the last 1000 timesteps. Final diversity and persistence are the number and the fraction, respectively, of the initial species whose biomass stayed above the extinction threshold throughout the simulation. Biomass abundance, productivity, and consumption are calculated as summed totals for the whole ecosystem and/or each guild of species. Plant productivity is the rate of biomass increase due to growth minus loss due to rewards production. Rewards productivity is the rate of rewards production minus self-limitation. Animal productivity is the rate of biomass increase due to assimilation minus losses due to metabolic maintenance. Consumption is the rate of biomass assimilated by consumers divided by assimilation efficiency. Species-level variability for the whole ecosystem (Fig. 4e) is the averaged coefficients of variation of biomasses (CV = standard deviation/mean) of all surviving species in the ecosystem. Species-level variability for each guild (Fig. 6e) is the averaged CVs of all surviving species within that guild. Guild-level variability for each guild (Fig. 6f) is the CV of the summed biomass of all species in that guild. Guild-level variability of the whole ecosystem (Fig. 4f) is the averaged CVs for five guilds (all plants, herbivores, all added animals, omnivores, and carnivores), which standardizes the grouping of species into guilds across treatments. Ecosystem-level variability (not shown) is the CV of the summed biomass of all species in the ecosystem.

**Feedback control**. To disentangle effects of mutualistic feedbacks from effects of floral rewards, we ran multiplex simulations with mutualism "turned off" ("feedback control"), in which all feedbacks (dashed and purple arrows in Fig. 1) between vegetation, rewards, and pollinators are severed. This control transforms plants with pollinators into two independent biomass pools: a plant-without-pollinators (vegetation) pool and a rewards pool, both with constant production rates. In this way, rewards production is forced to match that of the multiplex model in the absence of mutualistic feedbacks even though these feedbacks also generated the production rate through dynamics over the course of the multiplex simulations. All feeding interactions (gray arrows) remain the same.

Specifically, we modified the dynamics of each former plant with pollinators $i$ so that its vegetative biomass follows the dynamics and parametrization of plants without pollinators (Eq. 4, $r_i = 0.8$) and its rewards biomass follows:

$$\frac{dR_i}{dt} = \left(\overline{\beta_i B_i - s_i R_i}\right) - \sum_{j \in \text{pollinators}} C_{ji}(R_i)/e_{ji} \tag{10}$$

with fixed production rate $\left(\overline{\beta_i B_i - s_i R_i}\right)$ equal to $i$'s average net rewards production during the last 1000 timesteps of the multiplex simulations. In this manner, vegetation is not dependent upon pollinator consumption of rewards nor on rewards production, and rewards production is fixed and not dependent upon vegetation. All other species followed the same dynamic equations and parameterization as in the multiplex simulations.

We applied these feedback controls to the four multiplex treatments (RO Low, RO High, RP Low, RP High) and initialized all species at biomass $B_i = 10$ and rewards nodes at $R_i = \overline{R_i}$, the average rewards biomass for each plant with pollinator species $i$ during the last 1000 timesteps of the multiplex simulations. Simulations were run for 5000 timesteps to approximate steady state (Supplementary Fig. 9). We compared the results of these simulations with those of the original multiplex simulations by measuring absolute differences in persistence and total biomass at timestep 5000, where the effect of feedback = (multiplex − control). To assess differences in these ecosystem metrics due to guilds, we calculated absolute differences in the fraction of persisting species composed by each guild:

$$\frac{\text{multiplex final guild diversity}}{\text{multiplex final diversity}} - \frac{\text{control final guild diversity}}{\text{control final diversity}} \tag{11}$$

and the fraction of ecosystem biomass composed by each guild:

$$\frac{\text{multiplex guild biomass}}{\text{multiplex total biomass}} - \frac{\text{control guild biomass}}{\text{control total biomass}}. \tag{12}$$

If these effects of feedbacks evaluate to positive numbers, feedbacks in multiplex simulations have a positive effect, i.e. they increase persistence or biomass of the ecosystem or guild. If effects of feedbacks evaluate to negative numbers, feedbacks decrease persistence or biomass. If, instead, effects of feedbacks evaluate to approximately zero, stability and function in our multiplex treatments can be attributed to the overall rates of plant (vegetative and rewards) productivity that emerge during those simulations.

**Reporting summary**. Further information on research design is available in the Nature Research Reporting Summary linked to this article.

## Data availability
Network structures and parameterization to reproduce Figs. 3–6 are available in the online repository at https://github.com/kayla-hale/Multiplex-Dynamics/.

## Code availability
Simulation and analysis code are available in the online repository at https://github.com/kayla-hale/Multiplex-Dynamics/.

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

## Acknowledgements

We thank Nicholas J. Kappler for critical insight on our dynamical model. This research was supported by National Science Foundation Graduate Research Fellowship DGE-1143953 to K.R.S.H., National Science Foundation grant DEB-1834497 to F.S.V., and Department of Energy grant DE-SC0016247 and National Science Foundation grants 1241253, 1313830, 1642894, 1754207, and 1934817 to N.D.M.

## Author contributions

K.R.S.H. and N.D.M. conceived of the study. K.R.S.H. and F.S.V. formulated the model and designed the simulations. K.R.S.H., F.S.V., and N.D.M. designed the analyses and wrote the manuscript. K.R.S.H. performed the simulations and analysis.

## Competing interests

The authors declare no competing interests.
