## [Peer Review File · Nature Communications]

Reviewers' Comments:

Reviewer #1:

Remarks to the Author:

The effect of mutualisms on the stability of complex systems has been long controversial. Classic theory claims that these positive interactions tend to destabilize systems, despite this type of interactions are rather frequent in nature and have been show to be crucial in many habitats to maintain biodiversity. Recent theory actually finds that mutualisms do stabilize systems under particular conditions. Until now, such positive interactions had not been 'truly merged' with other type of consumer-resource interactions (i.e. food webs) to better discern their effect on the multiplex networks. In this work, authors do synthesize both types of networks, by means of modelling, finding that the dynamics of mutualisms (pollination, in particular) not only does not destabilize systems but increases (1) stability, (2) diversity (species richness) and (3) several ecosystem functions. The work significantly advances the field by developing and applying mechanistic consumer-resource theory to multiplex ecological networks. Specifically, their multiplex network model integrates both the structure and dynamics of feeding and reproductive mechanisms by combining Williams and Martinez's niche model of food webs with Thébault and Fontaine's model of mutualistic networks. Network dynamics was further simulated by extending Brose et al.'s allometric network theory and incorporating Valdovinos et al.'s theory of the exchange of food for reproductive services between plants and their pollinators. In contrast to previous work, their main focus is on the mechanisms that dynamically generate effects of ecological interactions, i.e. production and consumption of trophic resources and reproductive services, which provide more realistic network architectures. I thus see the work as highly novel and I do think it will be of interest to many ecologists using a network approach to understand ecological interactions.

The ms is clearly written, references are updated, figures are all justified, and to the best of my knowledge (I must say that I do not have a strong background on ecological modelling) the analyses are correct. I also think the level of detailed provided in the methods is good enough so that other researchers could reproduce the models. The main key limitation of the study is also pointed out, i.e. inability to validate the model with empirical data, but this is well justified as unfortunately data are only available for very few systems).

Just some minor points:

- 1- Page 6, line 132. Where do the 238 plant-pollinator networks come from? Could you please clarify this?
- 2- Page 10, line 220. Please correct $CV = \text{standard deviation} / \text{mean}$ (not the opposite!)

Reviewer #2:

Remarks to the Author:

The authors apply consumer-resource models to understand the effect of mutualism in multiplex networks. They find that considering mutualistic interactions can increase diversity, stability and ecosystem functions of ecological communities. Although I am troubled by the lack of depth in the analysis of the model's behaviour and network structure, I did like the concepts and ideas presented in the manuscript. The major concerns that I had regarding the manuscript are listed below.

First and foremost, it is very difficult to disentangle the effects of the different components of the analysis. To understand the behaviour of the model, I would expect the authors to feed the model with a diverse set of network structures (modular, nested, random,...; Fontaine et. al., Ecology Letters

2011) and different parameterisations (i.e. include a sensitivity analysis of the parameters). Instead, the authors assume a set of values for the parameters and directly jump to the analysis of simulated network structures. Following this, it is challenging for me to determine whether the results found by the authors are a consequence of the structure of the multiplex network, the parametrization of the model, the nature of the model, or mechanisms that can be biologically interpreted.

Along similar lines, it seems to me that the basic difference between the FW treatment and the RO and RP treatments is that an additional resource is added (i.e. floral rewards). On the one hand, if I look at figure 4, I can consider the sum of all plants w/o pollinators, plants with pollinators and floral rewards as the overall resources of the system from the lower trophic level. The total biomass of these increases relative to the same in the FW counterpart, as the FW treatment only includes plants without pollinators in this trophic level (Figure 4b). If this is the case, is the observed increase in network stability surprising if consumers have additional resources to feed on? The authors work on this question in the "Mutualistic feedbacks" and "dynamical controls" sections of the results and methods, respectively. However, I struggled to see how the corresponding tests can be used as an adequate control (I found these sections to be particularly hard to follow, and I might have missed something). Indeed, I would like to see how figures 3 and 4 look when controlling for the overall resources from the lower trophic level. Most importantly, isn't the way the model adds the floral rewards equivalent to adding extra nodes to the networks in Figure 1 (i.e. nodes representing the floral rewards)? That is, it seems that adding floral rewards might not only change the model but also the structure of the networks, as it adds an extra set of nodes R_i . If I understood the manuscript correctly, these extra "nodes" were not explicitly considered when building the networks. Could the differences between the FW treatment and both RO and RP treatments be understood under this perspective?

Finally, the authors argue that studying the effects of mutualism on the stability of multiplex networks is necessary given the "contradictory conclusions" of previous theoretical work. The work they cite to illustrate this contradiction are studies focussing on the effects of network structure on community stability (May, Allesina and Sauve,...). However, the effects of network structure is never studied here. If we were to merge together two random networks---a random unipartite network and a random bipartite one---and run the same models, would the treatments FW, RO and RP showcase the same differences? Similarly, how would the results for RP change if we randomised the mutualistic part of the networks? If the different treatments showcased the same results for randomised multiplex networks, would the implications for network stability be the result of the way mutualistic species interact with each other or how these are characterised in the dynamical models?

Minor:

- Reward productivity is not properly introduced.
- L51: Multiplex network is a term that has never been properly introduced.
- L59: ``food web research has focused more on aquatic systems'. This might be a bit misleading. While one could say that there are more studies on aquatic food webs than there are studies of terrestrial food webs, I would not say that network ecology is short on the later. I also don't really understand how the two references (1942 and 2000) prove the point, as an extensive body of work has been produced on terrestrial food webs over the last 19 years.
- The authors repeatedly use the term "realistic" to describe simulated networks. As much as the models that they use to generate random networks try to mimic the structure of empirical networks, "realistic" is a subjective term that can be misleading. The appropriate term here is "simulated networks".
- The last paragraph of the intro sounds very much like methods.
- Figure 2 can go to the Supplementary.
- "Mutualistic feedbacks" and "dynamical controls" sections require **a lot** more hand-holding in order for them to be accessible to readers.

- I feel that there is a lot of information to digest in figures 3 and 4. I would suggest narrowing down the questions that each figure attempts to answer. Otherwise, it is very hard to grasp the overall message that each figure tries to convey. I believe that any information that is not essential for the overall conclusions of the manuscript should go to the supplementary information.
- Although the authors mention the limitations of their approach in the discussion, I struggled to overlook the somewhat arbitrary parametrisation of the model. For example, the authors focus on the effects of what I believe are two arbitrary values for the reward productivity. Why is a value of 0.2 low reward productivity? It is unclear to me if that value would be actually low in nature. Why not 0.1 or 0.01? I would like to see the behaviour of the same models along a "reward productivity" gradient.
- Similarly, how would do the authors reconcile the static representation that they use with the observed seasonal variability of plant-pollinator networks (Simanonok et. al., Ecosphere 2014)

Reviewer #3:

Remarks to the Author:

Review of "Pollinators in food webs: Mutualistic interactions increase diversity, stability, and function in multiplex networks" by Hale et al.

The authors study the effect of mutualistic interactions in ecological networks that also include consumer-resource interactions. They combine four previously developed ecological network models to simulate community dynamics. The four models are: (i) the stochastic niche model to generate realistic food web structures; (ii) a model by Thebault and Fontaine to generate realistic mutualistic network structures; (iii) the allometric trophic network model to describe energy flow and biomass dynamics; and (iv) a model by Valdovinos (one of the authors) to describe the exchange of food for reproductive services between plants and pollinators. The authors consider three cases: (i) food web, FW---only consumer-resource feeding interactions are included and only herbivores and omnivores are added in simulations; (ii) rewards only, RO---built on FW, pollinators that facilitate plant reproduction are also included but they only consume plant resources called "rewards" and only pollinators are added in simulations; and (iii) rewards plus, RP---built on RO, pollinators are allowed to engage in herbivory and omnivory and, as with RO, only pollinators are added in simulations. Based on their simulations, the authors conclude that mutualistic interactions between plants and pollinators can increase the diversity (number of species in a community), stability (measured as the coefficient of variation in biomass), and persistence (number of initial species remaining at the end of simulations) of ecological communities.

The idea of modelling more than one type of biotic interaction using a network approach is important and timely, although, as the authors readily admit, they are not the first to do so. Nevertheless, their combination of the four aforementioned ecological network models is novel and interesting. While it is straightforward to understand their overall approach (i.e., they combine four models to simulate species population dynamics and then measure community-level properties at the end of simulations), it is very difficult to follow what is actually going beyond this superficial level. Specifically, to make sense of the results section, one really needs to have read the methods section, which defeats the purpose of the Nature Communications format. Especially for readers who are not experts in pollination networks, the manuscript needs clearer and more detailed definitions and descriptions of terms, particularly with respect to how more general concepts (e.g., mutualism, pollination visits, rewards) are implemented in modelling. Additionally, the results section would benefit from a more logical narrative structure within each subsection. I organise my thoughts into four major comments: (1) Be more precise in the introduction when discussing terms such as mutualism, facilitation, consumer-resource interactions, and reproduction; (2) Provide more details about the modelling procedure outside of the methods section; (3) Restructure the narrative of the results section and

improve clarity; and (4) Justify some of the seemingly arbitrary modelling decisions. Although I like the authors' general approach, as the manuscript stands it is difficult for me to fairly assess the significance of their findings.

(1) Be more precise in the introduction when discussing terms such as mutualism, facilitation, consumer-resource interactions, and reproduction

In the introduction, the authors discuss "mutualism" very generally when the manuscript requires greater nuance and more precise and detailed definitions. For example, it is important to clarify the distinctions between the following terms: a mutualistic interaction, an interaction in which both parties benefit, i.e., (+, +); a positive/facilitative/commensal interaction (e.g., increasing plant reproduction rates), an interaction in which one party benefits from the presence of another, i.e., (neutral, +); a consumer-resource interaction, an interaction in which one party benefits at the expense of another, i.e., (+, -); and an antagonistic interaction, an interaction in which one or both parties are negatively affected by the presence of the other party, i.e., (neutral, -) or (-, -). These definitions may not be exactly what the authors have in mind, but my point is they need to be very clear about how the reader should understand the terms that they are using, especially since the authors intend to combine some of these concepts. For example, the authors introduce mutualism with the implicit definition of (+, +), L42 to L46, but then cite works that focus on facilitation, L48. They then implicitly recast mutualism as an "integrated consumer-resource theory of feeding and reproductive mechanisms," L50. Before arguing that mutualistic interactions are a subclass of feeding interactions, L55. Later in the introduction, the authors return to describing a mutualistic interaction as one in which there is a net benefit to both partners, L96. This segueing between different interpretations of "mutualism" without precise definitions at each point makes it difficult for someone unfamiliar with the literature to follow the authors' logic.

The authors should also make it clear in the introduction that they are not only interested in how to *represent* a mutualistic interaction in models, but also how the *presence* of mutualistic interactions in a community affects species (e.g., non-pollinating herbivores, omnivores, and carnivores) not involved in mutualistic interactions---this is really what the authors are assessing in this work, but this focus is not signalled to the reader in the abstract or first five paragraphs of the introduction.

"Stability" is repeatedly used but with inconsistent meaning: L44; L85; L86; L100; L138; L145; L152; L296. A particularly egregious example is L296: "These increases of stability and function due to increases in diversity and mutualism are broadly consistent with empirical observations of the effects of increased pollinator diversity in blueberry, watermelon, and other agroecosystems. Though consistent with empirical observations, our results are largely inconsistent with classic theory showing that mutualism destabilizes ecological networks"---The authors not only use a different modelling approach to that used in classic theory (dynamical models versus random matrix theory), but also use a completely different definition of stability (coefficient of variation versus the largest eigenvalue of the community matrix). The authors then contradict themselves by stating that their theory is able to "reconcile classic theory" (L311).

(2) Provide more details about the modelling procedure outside of the methods section

The authors should provide at least some qualitative description of the four models being combined (L109 to L114), either at the end of the introduction or at the beginning of the results section. Otherwise, the reader has no intuition about where results are coming from and how to interpret them. Consider the sentence: "Within each multiplex network treatment, we varied the reward productivity of plants with pollinators by a factor of five between `Low' and `High' productivity"

(L140). Without reading the methods section, this sentence carries very little meaning to the reader--- for example, they aren't told until the methods section that "rewards" are actual biomass that is being produced by plants.

The authors also need to provide in a non-methods section some qualitative description of the community-level summary statistics (e.g., persistence, diversity, stability, biomass, productivity, consumption) that they measure. Otherwise, these words simply function as placeholders for abstract concepts that the reader has to juggle as they wade through the results section.

(3) Restructure the narrative of the results section and improve clarity

The results section is extremely difficult to follow. This is in part because the reader goes in with very little understanding of how the models are implemented and what summary statistics (e.g., persistence) actually describe (see major comment 2, above). But it is also difficult to follow because each subsection jumps around between results without a clear narrative structure. Results are seemingly plucked out of thin air; sometimes numbers are presented and sometimes they aren't; new, unfamiliar terms are introduced without definitions.

Including the full sets of results for each subsection in a table would help greatly. For example, the authors assert that "mutualism stabilizes population dynamics in multiplex treatments compared to FWs especially as diversity and mutualism increase, with the minor exception of the High RP treatment" (L235)---Being able to compare numerical results across models as the existing text does not provide enough support for this claim.

The authors should also add a paragraph explaining how they ensure fair comparisons can be made between FW, RO, and RP cases. The three cases differ in the number and type of interactions modelled and the numbers and types of species added (which impacts the amount of biomass added and the levels of productivity that are possible). For example, does adding a herbivore in the FW case add the same amount of starting biomass as adding a pollinator in the RO case? Do pollinators face the same negative effects from interactions as do herbivores?

"...these differences increased as larger fractions of the multiplex networks were directly engaged in mutualistic interactions" (L154)---How is a mutualistic interaction defined in the models?

"This difference is primarily due to the consistently high persistence of pollinators (~60%) compared to the persistence of added species in the FW treatments..." (L158)---But persistence measures the number of initial species remaining at the end of simulations (L556), so what does "persistence of added species" mean? In addition to this internal inconsistency, the reader needs the definition of persistence earlier, because otherwise sentences like "in all treatments, any decreases in persistence were not strong enough to prevent overall increased final diversity (Fig. 3b) with increased initial diversity and mutualism (Fig. 3a)" make little sense (L172; although honestly the entire sentence is ambiguous even with the definition of persistence in mind).

"... the increasing persistence of omnivores..." (L161)---Does "omnivores" include pollinators that also act as herbivores and omnivores?

What's an "RO FW" (L167)? What's an "(RO) FW" (L181)?

I think the authors meant to define the coefficient of variation as $CV = \text{standard deviation} / \text{mean}$ (L220). Even so, they need to explain which distribution the mean and standard deviation are calculated from. Is it variation across species, across simulations, across time? Otherwise, sentences

like "species on average were much more variable ($0.01 \leq CV \leq 0.03$), especially in Low rewards treatments where plants with pollinators and their rewards contributed large amounts of variability" provide little meaning.

The hypothesis the authors present is very difficult to follow, L250: "The control represents the hypothesis that the effects of pollinators on our systems are solely due to the total productivity of resources at the base of the food web that emerged due to pollination services during the original simulations." This is an important analysis, yet the control scenario appears to be presented as the actual working hypothesis rather than the simpler null hypothesis. "Productivity of resources" is not defined, and it is not described how this productivity emerges from "pollination services."

"On average, community species persistence..." (L253)---Average of what? What does "community species persistence" represent---this is a term only used once in the text?

What two-tailed test (L256) is being performed?

The phrase "minor exception" is used frequently (L171; L178; L237; L258; L283)---I would advise against using the subjective qualifier, "minor," as "exception" on its own is adequate. (Otherwise, the authors give the impression that they are favouring a particular outcome.)

"Former pollinators decreased in abundance despite access to up to twice the biomass previously available as rewards. In contrast, original herbivores increased in abundance in response to the increased vegetative biomass in the controls" (L264). This is an interesting and curious result that needs to be discussed further. Is it a direct consequence/artefact of modelling decisions or is it a surprising emergent property?

"More broadly, negative effects of mutualism consistently extend to none of our six other measures of ecosystem structure and function (diversity, persistence, abundance, productivity, consumption, guild-level stability) while positive effects of mutualism apply to all of our High rewards treatments and most of our Low rewards treatments" (L304). This sentence is very unclear---are the authors simply trying to say that, according to simulations, mutualism has a positive effect on the six measures?

New terms continue to appear in the discussion section, e.g., "omnivorous pollinators" (L331). Although I will add that the discussion is very nice from L358 onwards.

(4) Justify some of the seemingly arbitrary modelling decisions.

The methods section is generally well-written and comprehensive. There are, however, a few choices that should be justified to the reader.

Why were 102 food webs used (L129; i.e., why not a round 100 or 120)?

Why were there twice as many animal-pollinator species than plant-with-pollinator species (L417)?

Why does connectance decrease with species richness with RO (L433) but is kept constant with RP (L438)?

How do the authors confirm that simulations are at a dynamical steady state (L535)? Does steady state include the possibility that some of the initial species have gone extinct?

"One guild is just the rewards of all plants with pollinators" (L546)---Is this a guild of nectar?

Minor comments

-- L111. The authors actually use the stochastic niche model, so this version should be cited rather than or in addition to the original niche model.

-- 126. Why do the authors write "potentially antagonistic interactions"? Herbivory and carnivory are unambiguously antagonistic in that one individual is being negatively affected by the actions of another individual.

-- L185. Or just "compensated"?

Reviewer #1 (Remarks to the Author):

The effect of mutualisms on the stability of complex systems has been long controversial. Classic theory claims that these positive interactions tend to destabilize systems, despite this type of interactions are rather frequent in nature and have been show to be crucial in many habitats to maintain biodiversity. Recent theory actually finds that mutualisms do stabilize systems under particular conditions. Until now, such positive interactions had not been ‘truly merged’ with other type of consumer-resource interactions (i.e. food webs) to better discern their effect on the multiplex networks. In this work, authors do synthesize both types of networks, by means of modelling, finding that the dynamics of mutualisms (pollination, in particular) not only does not destabilize systems but increases (1) stability, (2) diversity (species richness) and (3) several ecosystem functions. The work significantly advances the field by developing and applying mechanistic consumer-resource theory to multiplex ecological networks. Specifically, their multiplex network model integrates both the structure and dynamics of feeding and reproductive mechanisms by combining Williams and Martinez’s niche model of food webs with Thébault and Fontaine’s model of mutualistic networks. Network dynamics was further simulated by extending Brose et al.’s allometric network theory and incorporating Valdovinos et al.’s theory of the exchange of food for reproductive services between plants and their pollinators. In contrast to previous work, their main focus is on the mechanisms that dynamically generate effects of ecological interactions, i.e. production and consumption of trophic resources and reproductive services, which provide more realistic network architectures. I thus see the work as highly novel and I do think it will be of interest to many ecologists using a network approach to understand ecological interactions.

- We thank the reviewer for this clear summary that succinctly characterizes the context, methods, findings, and significance of our work.

The ms is clearly written, references are updated, figures are all justified, and to the best of my knowledge (I must say that I do not have a strong background on ecological modelling) the analyses are correct. I also think the level of detailed provided in the methods is good enough so that other researchers could reproduce the models.

- We very much appreciate these comments.

The main key limitation of the study is also pointed out, i.e. inability to validate the model with empirical data, but this is well justified as unfortunately data are only available for very few systems). Just some minor points:

1. Page 6, line 132. Where do the 238 plant-pollinator networks come from? Could you please clarify this?

Response: We apologize for the lack of clarity. We had previously specified (original L111) that we generated them using Thébault and Fontaine’s model of mutualistic networks but failed to cite that reference later in the original manuscript. We now include that reference along with more details on **L135-137** as follows “238 plant-pollinator networks⁴⁹ (Fig. 2b) of varying species diversity ($S_p = A + P = 9, 12, \dots, 57$) and empirically-observed

pollinator-to-plant ratio ($A/P = 2$) and ranges of connectance and nestedness (S1).” In our Methods **L465-473**, we further elaborate on this description and provide a full technical description in the supplementary material (**S1**).

2. Page 10, line 220. Please correct CV= standard deviation / mean (not the opposite!)

Response: Thank you for noticing this; it has been corrected on **L249**.

Reviewer #2 (Remarks to the Author):

The authors apply consumer-resource models to understand the effect of mutualism in multiplex networks. They find that considering mutualistic interactions can increase diversity, stability and ecosystem functions of ecological communities.

- We agree with this summary.

Although I am troubled by the lack of depth in the analysis of the model's behaviour and network structure,

- We agree that further analysis would improve our manuscript. We follow this suggestion by applying additional analyses on the relative roles of mutualistic rewards and feedback dynamics between mutualistic partners (**L293-305**). We believe that with this added depth, our research and its extensive sensitivity analyses (**S2**) meets or exceeds the standards for such research established in the best journals.

I did like the concepts and ideas presented in the manuscript.

- We sincerely appreciate this compliment.

The major concerns that I had regarding the manuscript are listed below.

First and foremost, it is very difficult to disentangle the effects of the different components of the analysis.

- We agree that disentangling the effects is difficult, but we have extensively revised and clarified our manuscript to more clearly describe the different components of our analysis. These clarifications extend throughout the manuscript including overall organization, streamlining of sentence structure, and use of terms. While the effects themselves constitute the central findings of our work, we go further to show that the added productivity generated by mutualistic rewards is primarily responsible for the overall effects, while the dynamic feedbacks between mutualists lead to more subtle effects on the composition and abundance of species and guilds in our model ecosystems.

To understand the behaviour of the model, I would expect the authors to feed the model with a diverse set of network structures (modular, nested, random,...; Fontaine et. al., Ecology Letters 2011) and different parameterisations (i.e. include a sensitivity analysis of the parameters).

- We wholeheartedly agree with the importance of such sensitivity analyses and variation in network structures to understand the behavior of our model. To address this concern, we improved our descriptions of the large variation we had previously explored (**S1**) including our extensive sensitivity and uncertainty analyses (**S2**) and additionally added a set of “feedback control” treatments in which rewards nodes and their corresponding vegetation nodes are independent (**L294-302**). We are primarily interested in the effects of mutualism on ecologically-structured networks, as we believe this allows us to best

address a central concern, i.e., the effects of mutualism on existing ecosystems, and leads to the most relevant and accurate predictions on how variation of mutualism in nature may affect the observed behavior of ecosystems. We thus focused on hierarchical food webs and bipartite plant-pollinator networks, with mutualists occurring in adjacent trophic levels. In **S1** we describe that our simulated plant-pollinator networks span the wide ranges of connectance ($C_p = 0.09$ to 0.67 , **S1, Fig. S4**), nestedness ($\text{NODF}_{\text{st}} = -1.0$ to 1.1 , **S1**), and modularity ($Q_{\text{st}} = -0.34$ to 0.64 , **S1**) observed in empirical pollination networks within a computationally-tractable range of diversity ($S_p = 9$ to 57 , **L135-137**). By interconnecting these plant-pollinator networks with niche-model food webs of fixed complexity, our multiplex network structures span large ranges of species diversity ($S = 56$ to 88), connectance ($C = 0.04$ to 0.17 , **Fig. S3**), and mutualistic prevalence (from directly involving 16% to 65% of species, **L498-500**), with potentially highly nested, non-nested, or modular mutualistic components. Though this variation in networks structure does not include explicitly random networks, the variation we studied is of critical ecological importance and focuses our research on what we feel is most informative while leaving much further research for future studies.

Instead, the authors assume a set of values for the parameters and directly jump to the analysis of simulated network structures.

- We believe our > 24,000 networks each subjected to six treatments and (now) two feedback controls (for a total of almost 200,000 uniquely parameterized simulations, **L587-589**) cover a very large range of variability in network structure and parameterization. Indeed, our allometric parameterization of the feeding links in our networks *within* treatments relies upon consumer-resource body size ratios drawn from a lognormal distribution with mean 10 and standard deviation 100 (**L561-572**), such that the food web components of our networks can include interactions ranging from predator-prey to parasite-host. Furthermore, we used three of these network treatments in extensive sensitivity and uncertainty analyses (**Tables S2-3**) totaling 55 combinations of the free parameters in our model, i.e. those unconstrained by allometry. As such, “a set of values” appears to misrepresent the large range of network structures and model parameterizations investigated in our study. We clarified this point in the text (**L148-151**) as “Sensitivity and uncertainty analyses (S2) revealed a pivotal role of floral rewards in determining ecological effects of mutualism (Tables S1-S3, Fig. S5). We illustrate this role by presenting results from networks with High ($\beta = 1.0$), Low ($\beta = 0.2$), and no rewards productivity (Eqn. 4).”

Following this, it is challenging for me to determine whether the results found by the authors are a consequence of the structure of the multiplex network, the parametrization of the model, the nature of the model, or mechanisms that can be biologically interpreted.

- We agree that such understanding is a major challenge, though we believe we made major progress in addressing this challenge with our many network structures and parameter combinations. Our supplementary analyses indicate that the added productivity generated by mutualistic rewards is primarily responsible for our observed effects (**S2, S3**) with the dynamic feedbacks between mutualists instigating more subtle

effects on the distribution of species (**L306-319**). While there is certainly more to learn about the effects and causes we describe, we feel that our advance is novel and substantive and, as such, a significant contribution that will help guide productive advances in future work such as those described by the reviewer.

Along similar lines, it seems to me that the basic difference between the FW treatment and the RO and RP treatments is that an additional resource is added (i.e. floral rewards).

- We very much agree that floral rewards comprise a fundamental difference between our FW and multiplex treatments. However, it is not the only one. Another basic difference is the dependence of plant growth on the behavior of mutualistic partners. While the former could simply just provide more autotrophic biomass to an ecosystem, the latter adds interdependence that can destabilize systems due to positive feedbacks. It is the interplay of these forces that form the basic differences among treatments.

On the one hand, if I look at figure 4, I can consider the sum of all plants w/o pollinators, plants with pollinators and floral rewards as the overall resources of the system from the lower trophic level.

- We agree this would be a fair consideration that we now more extensively explore.

The total biomass of these increase relative to the same in the FW counterpart, as the FW treatment only includes plants without pollinators in this trophic level (Figure 4b).

- This is true. However, though we introduced rewards, we also introduced costs of producing them (see **Fig. 1** and κ in **Table S1**). That the total biomass from the lower trophic levels at the *end* of simulations was higher than in FWs (now **Fig. 6b**) and that it furthermore enriched the upper trophic levels in High rewards productivity treatments was a result that emerged from our dynamic equations over the course of simulations, as we now describe on **L294-300**. We consider the importance of floral rewards biomass in a food-web context to be a critical result of our model, as this resource and its dynamics are not typically accounted for in food webs, though it is increasingly garnering attention in the pollination literature (e.g. Valdovinos *et al.* 2016, Baude *et al.* 2016).

If this is the case, is the observed increase in network stability surprising if consumers have additional resources to feed on?

- It may not be surprising, but it is also not necessarily obvious. Our model is a multispecies extension of the Rosenzweig-MacArthur model, the model from which the famous “paradox of enrichment” emerged wherein additional resources destabilize consumer-resource interactions. This paradox provides a mechanism that could be responsible for what May (1981) called a destabilizing “orgy of mutual benefaction.” As such, we feel it to be a substantial advance to show that such instabilities due to mutualism do not arise in the vast majority of our multiplex networks even at high reward productivities. It is also a topic that deserves more attention in future theoretical and applied work.

The authors work on this question in the "Mutualistic feedbacks" and "dynamical controls" sections of the results and methods, respectively. However, I struggled to see how the corresponding tests can be used as an adequate control (I found these sections to be particularly hard to follow, and I might have missed something). Indeed, I would like to see how figures 3 and 4 look when controlling for the overall resources from the lower trophic level.

- We thank the reviewer for pointing this out and we have worked to address this concern thoroughly including by creating a new class of “feedback controls,” following the reviewer’s suggestions (below). In particular, supplementary **Fig. S7a-b** in the style of the figures now numbered 5 and 6 shows visually that the feedback controls account for overall abundance of resources in the lower trophic levels. Results for the original dynamical control (which we now call the “rewards partitioning control”) are also presented in the supplementary information (**Fig S10a-b**).

Most importantly, isn't the way the model adds the floral rewards equivalent to adding extra nodes to the networks in Figure 1 (i.e. nodes representing the floral rewards)? That is, it seems that adding floral rewards might not only change the model but also the structure of the networks, as it adds an extra set of nodes R_i . If I understood the manuscript correctly, these extra "nodes" were not explicitly considered when building the networks. Could the differences between the FW treatment and both RO and RP treatments be understood under this perspective?

- Floral rewards are indeed modeled as additional nodes in our networks but their dynamics are far from equivalent to the nodes in our networks which represent populations of plants and animals (**L228-233**). For example, the productive potential of rewards is largely independent of their abundance and rewards have negligible maintenance costs. However, the reviewer’s comments provide an excellent suggestion for clarifying this point with a feedback control that address the reviewer’s (and several of our own) concerns. That is, we substituted the “dynamical controls” previously in the main text with “feedback controls” presented in the Results **L293-319**, in the Methods **L638-673**, and visually in **Fig. S6b**. These controls retain the extra set of rewards nodes and forces their productivity to a constant value equal to the reward production rates seen in the multiplex networks, but turns off the mutualistic feedbacks that interconnect the dynamics of rewards, vegetation, and pollinators. In particular, the control preserves resource partitioning between rewards and vegetation, which may be an important feature for coexistence. Similar to our previous “dynamical control” results (now clarified and included in **S3**), we see that food webs with added floral rewards nodes and productivity have broadly similar diversity, persistence, biomass, and productivity as in the multiplex simulations but differing community composition in terms of guild persistence and biomass (**L308-312**, **Fig. S7-8**). In summary, we show how many but not all of our results do come from adding rewards nodes as the reviewer suggests. Though we highlighted the importance of rewards in our previous manuscript (now **L412-431**), we clarified and expanded upon now they shape our results in an additional paragraph in the Discussion (**L351-366**).

Finally, the authors argue that studying the effects of mutualism on the stability of multiplex networks is necessary given the "contradictory conclusions" of previous theoretical work. The work they cite to illustrate this contradiction are studies focussing on the effects of network structure on community stability (May, Allesina and Sauve,...). However, the effects of network structure is never studied here.

- Though these works focus on different issues than our work such as the behavior of more random vs. more structured networks, they also highlight the similar issue of how the prevalence of mutualistic interactions affects ecological networks. We believe the findings from those studies (e.g., mutualism is destabilizing) are central to ecologists' understanding of the effects of mutualism on ecosystems, in part because relatively little theoretical and empirical work on this topic exists (Kéfi *et al.* 2017). As such, we continue to discuss the findings of our study in relation to this previous work, but also have included more specific justification of how our work, which focuses on dynamical mechanisms, is an important philosophical departure from the more static approaches of previous theory (L72-85, L92-103, L367-377). Additionally, we did study effects of network structure previously but did not highlight them due to their small size compared to the effects we did highlight. We have now corrected this oversight in S2 by more clearly and quantitatively describing the effects of network structure on our results.

If we were to merge together two random networks---a random unipartite network and a random bipartite one---and run the same models, would the treatments FW, RO and RP showcase the same differences? Similarly, how would the results for RP change if we randomised the mutualistic part of the networks? If the different treatments showcased the same results for randomised multiplex networks, would the implications for network stability be the result of the way mutualistic species interact with each other or how these are characterised in the dynamical models?

Response:

These are very interesting and important questions, but we leave their answers for future studies to address and instead focus our analyses on more ecologically structured networks as described above. Potential answers include effects contrary to ours with mutualism being broadly destabilizing and/or reducing the function of randomized networks. Such a result would suggest that our observed effects of mutualism are restricted to more realistically-structured networks. This would be coherent with previous work which has shown that the overall stability of food webs and mutualistic networks appears to emerge from observed patterns in their structure such as consumer-resource body size ratios and nestedness. On the other hand, our findings that key structural properties such as diversity, connectance, nestedness and modularity explain far less variation in network behavior than does reward productivity (S2) could suggest that our finding extends at least qualitatively to more randomized network structures. This would suggest that mechanisms of mutualisms, especially food provisioning by mutualists, more generally stabilize and increase function in networks such as those discussed in L453-456.

More importantly, it would be unfortunate if our exclusion of more randomized networks delayed what we think are seminal findings regarding how mutualism affects ecosystems, using some of the best available knowledge on food webs and mutualistic networks in a synthesis of their structure, dynamics, and parameterizations. We structured trophic and mutualistic

interactions similar to those seen in nature and we parameterized the species according to empirically observed body size ratios and their associated metabolic rates. A key benefit of this focus is our ability to more directly compare our results with effects of mutualism seen in the field (L340-342, L389-403); we assume empirical networks to be more similar to our multiplex networks than to random networks (L405-412). Though it would certainly be interesting to explore how mutualism affects more randomly and otherwise structured and parameterized multiplex networks, it is beyond the scope of our research which we believe is already quite large and of central scientific interest.

Overall, we consider our results of increased ecosystem stability and function in the presence of additional resources and mutualistic feedbacks to be an important contribution to the current ecological network literature. Classic theory suggests that the paradox of enrichment, increased diversity, and mutualistic feedbacks should all “destabilize” ecological systems. Our results instead suggest the opposite and in fact intuitively result in *increased stability* when modeling patterns of network structure and parameters observed in nature plus biological mechanisms considered stabilizing within mutualism research.

Minor:

1. Reward productivity is not properly introduced.

Response: We revised our text to explicitly introduce the term in the Results section on L148-151: “Sensitivity and uncertainty analyses (S2) revealed a pivotal role of floral rewards in determining ecological effects of mutualism (Tables S1-S3, Fig. S5). We illustrate this role by presenting results from networks with High ($\beta = 1.0$), Low ($\beta = 0.2$), and no rewards productivity (Eqn. 4).” and L169-171: “The *intensity* of mutualism in our treatments varies with rewards productivity (Table 1) from high (High RO, High RP) to low (Low RO, Low RP) to none (RO FW, RP FW).” We also introduced **Table 1** as reference for terms such as rewards productivity that are used throughout the text.

2. L51: Multiplex network is a term that has never been properly introduced.

Response: This was a critical point for us to address. We now introduce “multiplex networks” in L47-50 as “Here, we address such disparities between theory and observation by developing and applying consumer-resource theory of feeding and reproductive mechanisms that integrates food webs and mutualistic networks into “multiplex” networks containing different types of interactions.” It is also defined in **Table 1**.

3. L59: “food web research has focused more on aquatic systems”. This might be a bit misleading. While one could say that there are more studies on aquatic food webs than there are studies of terrestrial food webs, I would not say that network ecology is short on the later. I also don't really understand how the two references (1942 and 2000) prove the point, as an extensive body of work has been produced on terrestrial food webs over the last 19 years.

Response: Though we acknowledge that much work has been done in recent years on terrestrial food webs, we now include a recent citation (L58-59) that addresses this point

directly: in a database of 290 natural ecosystems, Brose *et al.* (2019) recorded 189 aquatic food webs compared to 73 terrestrial food webs, of which only 22 were in aboveground ecosystems. We believe that the relatively limited data in terrestrial aboveground ecosystems has hampered previous attempts at using mechanism-based approaches to synthesizing multiple interaction types into multiplex ecological networks.

4. The authors repeatedly use the term "realistic" to describe simulated networks. As much as the models that they use to generate random networks try to mimic the structure of empirical networks, "realistic" is a subjective term that can be misleading. The appropriate term here is "simulated networks".

Response: We removed questionable uses of the word “realistic” but retain its use in the manuscript to emphasize the distinction between more and less realistic (e.g., more randomized) networks. Our networks integrate trophic and mutualistic interactions with what we believe to be realistic variation in both structure of the interactions (**S1**) and in parameterization of vital rates determined by body size and the consumer-resource mechanisms generating these dynamics (Methods: **Parameterization**). Only calling our networks “simulated” would distract from this central focus our research.

5. The last paragraph of the intro sounds very much like methods.

Response: We agree. Much of the work of this project was in synthesizing the multiplex model given the different biases in the ecological sub-disciplines of mutualism and food web research. To highlight this point and improve readability, we moved and expanded upon that content in a first section of the Results called the “**The multiplex model**” (**L119-179**). The main points are illustrated in **Figs. 1-2**.

6. Figure 2 can go to the Supplementary.

Response: We retain the figure (now numbered 3) in the main text because we believe that providing visual examples of our simulations, which are ultimately the raw data upon which we built our conclusions, improves the accessibility of our Methods and Results.

7. "Mutualistic feedbacks" and "dynamical controls" sections require *a lot* more hand-holding in order for them to be accessible to readers.

Response: We thank the reviewer for drawing our attention to this point. We replaced the dynamic control with the recommended “feedback control,” which better addresses the question of the effect of mutualistic dynamics versus primary productivity in our multiplex model. Due to space constraints, we limited the content we presented in the Results, but to increase accessibility we clarified our conceptual framework in **L294-305**, added a visual illustration of the multiplex model (**Fig. 1**) from which the feedback control is clearly derived by removing the blue arrows (**L300-301**; it is also presented directly in **Fig. S6b**), and clearly stated the interpretations of potential outcomes in the Methods on **L661-674**. Finally, we added detailed results to the supplementary material (**S3**) including figures in the style of the main text results (**Fig. S7-8**).

8. I feel that there is a lot of information to digest in figures 3 and 4. I would suggest narrowing down the questions that each figure attempts to answer. Otherwise, it is very hard to grasp the overall message that each figure tries to convey. I believe that any information that is not essential for the overall conclusions of the manuscript should go to the supplementary information.

Response: We greatly appreciate this point and include a new figure (now numbered 4) that summarizes the broader conclusions of our study before the more detailed figures now numbered 5 and 6. We believe this helps readers more easily digest the information in Figs. 5 and 6. We also considered rewriting the main text to show only results for networks of intermediate initial diversity (e.g. $S = 72$) and referring readers to the supplementary information for detailed results across increasing diversity or providing only averages across all networks (as now shown in Fig. 3). However, this would de-emphasize that multiple dimensions of stability and function are affected by both the presence *and* the prevalence of mutualism. We believe these patterns are best illustrated visually through detailed figures.

9. Although the authors mention the limitations of their approach in the discussion, I struggled to overlook the somewhat arbitrary parametrisation of the model. For example, the authors focus on the effects of what I believe are two arbitrary values for the reward productivity. Why is a value of 0.2 low reward productivity? It is unclear to me if that value would be actually low in nature. Why not 0.1 or 0.01? I would like to see the behaviour of the same models along a "reward productivity" gradient.

Response: This excellent suggestion helped us reframe our study to show that we actually do investigate a "rewards productivity gradient." That is, our study does characterize results from no reward productivity in food webs, to low productivity in multiplex networks of 0.2, to high productivity in multiplex networks of 1. We also clarify that our parameter choice, as presented in the main text, was the result of additional analyses (L148-151). We had previously reported persistence in our treatments across rewards productivities of 0.1 to 1.4 in **Tables S1-S2**, but have now also depicted those results visually (**Fig. S5**). In particular, 0.1 was the lowest value we explored because at that level all animals went extinct in the large majority of simulations, especially in RO networks where food limitation for pollinators is stark. We used 0.2 for low productivity in the main text because it similarly limited animals in many cases but also allowed for variation in outcomes. More generally, it is also unclear to us what a "low" or "high" parameter value would be in nature. We hope this work will draw attention to such topics, as discussed in **L435-454**.

10. Similarly, how would do the authors reconcile the static representation that they use with the observed seasonal variability of plant-pollinator networks (Simanonok et. al., Ecosphere 2014)

Response: This question is relevant to many theoretical studies such as ours in which the dynamics of seasonal variability are not explicitly addressed. We hope work on this topic

will following the trajectory of predictive theory in aquatic food webs: here we propose a mechanistic model for ecological dynamics similarly to that of Brose *et al.* 2006, which we hope to use in the future with an empirical system where we can test the ability of increasingly mechanistic models to reproduce observed seasonal variability through emergent ecological dynamics similarly to Boit *et al.* 2012. We now touch on this critical point on **L417-420** as: “accounting for seasonality of reward production and pollinator activity could greatly decrease estimates of resource demand needed to sustain pollinators. This highlights the need for improved theory of seasonal effects on both food webs¹⁸ and mutualistic networks⁶⁴.”

Reviewer #3 (Remarks to the Author):

The authors study the effect of mutualistic interactions in ecological networks that also include consumer-resource interactions. They combine four previously developed ecological network models to simulate community dynamics. The four models are: (i) the stochastic niche model to generate realistic food web structures; (ii) a model by Thebault and Fontaine to generate realistic mutualistic network structures; (iii) the allometric trophic network model to describe energy flow and biomass dynamics; and (iv) a model by Valdovinos (one of the authors) to describe the exchange of food for reproductive services between plants and pollinators. The authors consider three cases: (i) food web, FW---only consumer-resource feeding interactions are included and only herbivores and omnivores are added in simulations; (ii) rewards only, RO---built on FW, pollinators that facilitate plant reproduction are also included but they only consume plant resources called "rewards" and only pollinators are added in simulations; and (iii) rewards plus, RP---built on RO, pollinators are allowed to engage in herbivory and omnivory and, as with RO, only pollinators are added in simulations.

- We thank the reviewer for this concise and attentive summary of our research. While we had previously described our work as *three* cases (FW, RO, and RP), we now refer to our *six* treatments to clarify our spectrum from High to Low rewards productivity which we apply to both Rewards Only (RO) and Rewards Plus (RP) multiplex networks and to comparable FW networks with no rewards production. All mutualists in multiplex networks are converted into non-mutualists in the two FW cases as illustrated in **Fig. 2d-e**. We substantially restructured our text to clarify our approach in this respect (**L144-162**). We hope that this addition, as well as many others that were very helpfully recommended by this reviewer, substantially increase the comprehensibility of our work.

Based on their simulations, the authors conclude that mutualistic interactions between plants and pollinators can increase the diversity (number of species in a community), stability (measured as the coefficient of variation in biomass), and persistence (number of initial species remaining at the end of simulations) of ecological communities.

- This is largely true but we now further clarify that while mutualism does increase *diversity* (not necessarily persistence) in terms of the "*number* of initial species remaining at the end of simulations," *persistence* in terms of the *fraction* of initial species surviving to the end of the simulations increases in only one of our six treatments. We now explicitly define these terms in **Table 1**.

The idea of modelling more than one type of biotic interaction using a network approach is important and timely, although, as the authors readily admit, they are not the first to do so. Nevertheless, their combination of the four aforementioned ecological network models is novel and interesting. While it is straightforward to understand their overall approach (i.e., they combine four models to simulate species population dynamics and then measure community-level properties at the end of simulations), it is very difficult to follow what is actually going beyond this superficial level. Specifically, to make sense of the results section, one really needs to have read the methods section, which defeats the purpose of the Nature Communications format.

- We very much agree with this advice and now treat our models more as the results of our theoretical work integrating the structure and dynamics of trophic and reproductive interactions into multiplex networks. This emphasizes that a key part of our work is synthesizing trophic and reproductive interactions while accounting for differing concepts, techniques, and biases in the ecological sub-disciplines of mutualism and food web research. To clarify this, we now begin our Results with a section entitled “**The multiplex model**” (L119-179), in which we present our theoretical advancements and justify our approach. This better complies with the format of *Nature Communications* including by respecting the length limit of articles and improves readers’ ability to understand results using our model.

Especially for readers who are not experts in pollination networks, the manuscript needs clearer and more detailed definitions and descriptions of terms, particularly with respect to how more general concepts (e.g., mutualism, pollination visits, rewards) are implemented in modelling.

- We agree and now discuss these general concepts more explicitly in terms of ++ and other combinations of interactions in the Introduction on L54-58, 65-68, 72-85, 104-110, and pollination mutualisms specifically in Fig. 1 and Table 1.

Additionally, the results section would benefit from a more logical narrative structure within each subsection.

- We agree with and follow this advice. After presenting our model, we now outline a more logical narrative structure on L175-179 and then follow it. That outline reads, “We use these results to compare multiple measures of stability and function among treatments averaged over all initial diversity classes (Fig. 4) or within initial diversity classes (Figs. 5-6) at the species, guild, and ecosystem levels (Table 1). Below, we describe the effects of the presence, prevalence, and intensity of mutualism first on biodiversity, then on ecosystem function, and finally on temporal stability.”

I organise my thoughts into four major comments: (1) Be more precise in the introduction when discussing terms such as mutualism, facilitation, consumer-resource interactions, and reproduction; (2) Provide more details about the modelling procedure outside of the methods section; (3) Restructure the narrative of the results section and improve clarity; and (4) Justify some of the seemingly arbitrary modelling decisions. Although I like the authors' general approach, as the manuscript stands it is difficult for me to fairly assess the significance of their findings.

- These detailed recommendations on writing and organization were invaluable. We substantially restructured the Results section and rewrote the second half of the Introduction and the first half of the Discussion to emphasize clarity and precision in our definitions and modelling framework. We believe these revisions will highlight the comprehensiveness of our approach to understanding the effects of pollination mutualism on ecological networks. We respond point-by-point below.

(1) Be more precise in the introduction when discussing terms such as mutualism, facilitation, consumer-resource interactions, and reproduction

- In the introduction, the authors discuss "mutualism" very generally when the manuscript requires greater nuance and more precise and detailed definitions. For example, it is important to clarify the distinctions between the following terms: a mutualistic interaction, an interaction in which both parties benefit, i.e., (+, +); a positive/facilitative/commensal interaction (e.g., increasing plant reproduction rates), an interaction in which one party benefits from the presence of another, i.e., (neutral, +); a consumer-resource interaction, an interaction in which one party benefits at the expense of another, i.e., (+, -); and an antagonistic interaction, an interaction in which one or both parties are negatively affected by the presence of the other party, i.e., (neutral, -) or (-, -). These definitions may not be exactly what the authors have in mind, but my point is they need to be very clear about how the reader should understand the terms that they are using, especially since the authors intend to combine some of these concepts. For example, the authors introduce mutualism with the implicit definition of (+, +), L42 to L46, but then cite works that focus on facilitation, L48. They then implicitly recast mutualism as an "integrated consumer-resource theory of feeding and reproductive mechanisms," L50. Before arguing that mutualistic interactions are a subclass of feeding interactions, L55. Later in the introduction, the authors return to describing a mutualistic interaction as one in which there is a net benefit to both partners, L96. This segueing between different interpretations of "mutualism" without precise definitions at each point makes it difficult for someone unfamiliar with the literature to follow the authors' logic.

Response: We agree that our manuscript needs to more carefully define terms and concepts to better clarify our research and to make it easier to understand for the reader. There is a rather large distinction in the literature between defining mutualistic and other interactions in terms of their direct effects of one species on another (e.g. -, +, 0) and in terms of the processes (e.g. feeding, reproduction, etc.) that species directly participate in from which these effects emerge. Our work focuses on the processes and we agree we need to clarify how our work relates to the former "effects-based" research. We do this by following the reviewers advice and describing the discrepancies between the definition of mutualism as a strictly (+, +) effect-based interaction typically invoked in community ecology and a more dynamic net effect that emerges from a balance of "costs" and "benefits" involving feeding and reproductive services. We clarified our language to represent this distinction in **L54-58, L65-70** and explicitly discussed the core issue—classifying and modelling interactions based on *effects* versus *mechanisms*—in **L72-89**.

- The authors should also make it clear in the introduction that they are not only interested in how to *represent* a mutualistic interaction in models, but also how the *presence* of mutualistic interactions in a community affects species (e.g., non-pollinating herbivores, omnivores, and carnivores) not involved in mutualistic interactions---this is really what the authors are assessing in this work, but this focus is not signalled to the reader in the abstract or first five paragraphs of the introduction.

Response: We appreciate this advice to better clarify the scope of our research. We now describe that our scope is not only to understand how mutualism affects non-mutualistic

species, but also to understand how mutualistic species affect species more generally as well as guilds and whole ecosystems. We better signaled this by modifying the **Abstract (L26-31)** as follows: “Here, we synthesize both types of consumer-resource interactions to better understand the controversial effects of mutualism on ecosystems at the species, guild, and whole-community levels. We find that consumer-resource mechanisms underlying plant-pollinator mutualisms can increase the persistence, productivity, abundance, and temporal stability of both mutualists and non-mutualists in food webs.”

- "Stability" is repeatedly used but with inconsistent meaning: L44; L85; L86; L100; L138; L145; L152; L296. A particularly egregious example is L296: "These increases of stability and function due to increases in diversity and mutualism are broadly consistent with empirical observations of the effects of increased pollinator diversity in blueberry, watermelon, and other agroecosystems. Though consistent with empirical observations, our results are largely inconsistent with classic theory showing that mutualism destabilizes ecological networks"---The authors not only use a different modelling approach to that used in classic theory (dynamical models versus random matrix theory), but also use a completely different definition of stability (coefficient of variation versus the largest eigenvalue of the community matrix). The authors then contradict themselves by stating that their theory is able to "reconcile classic theory" (L311).

Response: This is a critical point and we appreciate the correction. We included **Table 1** to specify our definition of stability and our quantification of it by multiple metrics (see Table 1). We also removed our assertion of “reconcil[ing] classic theory” and revised our language when referencing previous theory to clarify definitional differences, e.g. **L92-101, L340-350, L379-388**, while also highlighting how this previous work has informed intuition about the effects of mutualism in complex networks on **L342-344**.

- (2) Provide more details about the modelling procedure outside of the methods section
- The authors should provide at least some qualitative description of the four models being combined (L109 to L114), either at the end of the introduction or at the beginning of the results section. Otherwise, the reader has no intuition about where results are coming from and how to interpret them. Consider the sentence: "Within each multiplex network treatment, we varied the reward productivity of plants with pollinators by a factor of five between 'Low' and 'High' productivity" (L140). Without reading the methods section, this sentence carries very little meaning to the reader---for example, they aren't told until the methods section that "rewards" are actual biomass that is being produced by plants.

Response: This is very helpful advice which we did our best to follow. To do this, we introduced a section to the Results entitled “The multiplex model” describing our network dynamics models (**L120-132, Fig. 1**), our network structure treatments (**L133-171, Fig. 2**), and justifying our parameterization (**L148-151**). **Table 1** includes referenceable definitions of the model treatments and outputs. We removed most of the methods overview from the Introduction.

- The authors also need to provide in a non-methods section some qualitative description of the community-level summary statistics (e.g., persistence, diversity, stability, biomass,

productivity, consumption) that they measure. Otherwise, these words simply function as placeholders for abstract concepts that the reader has to juggle as they wade through the results section.

Response: We recognize that our work is unusually comprehensive and needs improved clarification of our several measures of ecosystem stability and function. We followed this advice by including detailed definitions of our summary statistics, network treatments, etc. in **Table 1**.

- (3) Restructure the narrative of the results section and improve clarity
- The results section is extremely difficult to follow. This is in part because the reader goes in with very little understanding of how the models are implemented and what summary statistics (e.g., persistence) actually describe (see major comment 2, above). But it is also difficult to follow because each subsection jumps around between results without a clear narrative structure. Results are seemingly plucked out of thin air; sometimes numbers are presented and sometimes they aren't; new, unfamiliar terms are introduced without definitions.

Response: We substantially restructured the Results given this feedback. In particular, we defined summary statistics in **Table 1**, summarized all outputs in **Fig. 4**, and described the modeling framework in a first section of the Results called “**The multiplex model.**” At the end of this section (**L175-179**), we laid out a more logical narrative structure for the remainder of the results. Then, we structured our writing so that we refer to Fig. 4 *exclusively* when *averages* for each treatment were being compared, i.e. when reporting the effect of the *presence* and *intensity* of mutualism as defined by rewards productivity, or to **Figs. 5-6** *exclusively* when *patterns across increasing diversity/prevalence mutualism* or when *guild-level outputs* for each treatment are being compared. Finally, we presented numbers in the text only when they were not available in Figs. 4-6, i.e. when guild abundance, guild productivity, or within-guild variability are being compared across treatments.

- Including the full sets of results for each subsection in a table would help greatly. For example, the authors assert that "mutualism stabilizes population dynamics in multiplex treatments compared to FWs especially as diversity and mutualism increase, with the minor exception of the High RP treatment" (L235)---Being able to compare numerical results across models as the existing text does not provide enough support for this claim.

Response: Thank you for this simple but very helpful suggestion. We followed this advice by adding a new **Fig. 4** to facilitate numerical comparisons as suggested.

- The authors should also add a paragraph explaining how they ensure fair comparisons can be made between FW, RO, and RP cases. The three cases different in the number and type of interactions modelled and the numbers and types of species added (which impacts the amount of biomass added and the levels of productivity that are possible). For example, does adding a herbivore in the FW case add the same amount of starting biomass as adding a

pollinator in the RO case? Do pollinators face the same negative effects from interactions as do herbivores?

Response: We appreciate these important questions and agree that readers should have more readily available answers. Therefore, we revised our results to emphasize the comparability between different treatments due to species participating in as similar interactions as possible across treatments. We provided such clarification in several places. The new Results section, **The multiplex model**, begins these clarifications on **L151-179**. In particular, on **L144-147** we state that “As such, RO and RP treatments generate two different topological classes of multiplex networks for which we generate two groups of topologically-comparable food webs (RO FW and RP FW), described below.” We then conclude the following paragraph on **L160-162** with “These two Food Web treatments (RO FW, RP FW) control for network structure, including the varying numbers and trophic levels of species and links in RO and RP networks, to help elucidate the effects of mutualism in multiplex networks.” This section clarifies that added animals (herbivores or omnivores in FWs or pollinators in multiplex networks) are given the same links (including predation links from carnivory) in both FW treatments and multiplex networks. Both herbivores and pollinators are initialized with a starting biomass of 10 (**L171-175**). Both the vegetation and rewards of plants with pollinators are initialized to 10 as well (**L171-175**), so that the food availability for added animals is the same at the beginning of simulations, but the plants with pollinators have twice as much biomass initially as plants without pollinators. This added biomass is explicitly controlled for in the “feedback controls” section (**L300-302**) recommended by Reviewer 2.

- "...these differences increased as larger fractions of the multiplex networks were directly engaged in mutualistic interactions" (L154)---How is a mutualistic interaction defined in the models?

Response: We agree that this definition was previously problematically opaque to the reader. We addressed this extensively in our introduction by describing the difference between frequently invoked “effects-based” and our process-based definitions (**L72-85**), visually in **Fig. 1**, as a description in the **The multiplex model** section of the Results in **L120-131**, and as a short definition in **Table 1**: “A pollination link or mutualistic interaction between pollinator i and plant w/ pollinator j describes both the consumption of j ’s floral rewards by i and the reproductive services provisioning to the vegetative growth rate of j by i (Fig. 1). In the FW treatments, pollination links are switched to links in which i consumes the vegetative biomass of j , i.e. to herbivory links.”

- "This difference is primarily due to the consistently high persistence of pollinators (~60%) compared to the persistence of added species in the FW treatments..." (L158)---But persistence measures the number of initial species remaining at the end of simulations (L556), so what does "persistence of added species" mean? In addition to this internal inconsistency, the reader needs the definition of persistence earlier, because otherwise sentences like "in all treatments, any decreases in persistence were not strong enough to prevent overall increased final diversity (Fig. 3b) with increased initial diversity and mutualism (Fig. 3a)" make little sense (L172; although honestly the entire sentence is ambiguous even with the definition of persistence in mind).

- **Response:** This comment appears to result from a confusion between diversity in terms of a *number* of species and persistence in terms of a *fraction* of species where one can increase while the other can decrease as described in our second response to Reviewer 3 above. Instead of “persistence measures the number of initial species remaining at the end of simulations” as stated by the reviewer, L556 of the previous manuscript stated that, “We calculated species persistence as the fraction of the initial species that survived to the end of the simulation (i.e. whose biomass stayed above the extinction threshold).” This was also restated on L849-850 in the legend of the original Fig. 3 that presented persistence of our networks as “Persistence, the fraction of the initial diversity that persists to the end of the simulations, is shown...” We understand that this was insufficient and now better clarify this definition earlier on as suggested in Table 1 as well as in the Methods (L619-621) and legend of Fig. 5 (L943-945).

- "... the increasing persistence of omnivores..." (L161)---Does "omnivores" include pollinators that also act as herbivores and omnivores?

Response: For increased clarity on this point, we now define all guilds in **Table 1**. Then, in the Results section, we reserve the term “omnivore” to mean only those as defined in Table 1 and always specify if we include pollinators; we use the same strategy for the term “herbivore.” References to specific panels in Fig. 5 clarify this also, since the panels present guilds as defined in Table 1.

- What's an "RO FW" (L167)? What's an "(RO) FW" (L181)?

Response: We apologize for this previous lack of clarity and now improve our descriptions of these critically important treatments. This is now clearly defined in **Table 1** and the Results: **The multiplex model** section on L153-162. Additional clarifications are described by our response to Reviewer 3’s suggestions to describe how we “ensure fair comparisons can be made between FW, RO, and RP cases” above.

- I think the authors meant to define the coefficient of variation as $CV = \text{standard deviation} / \text{mean}$ (L220). Even so, they need to explain which distribution the mean and standard deviation are calculated from. Is it variation across species, across simulations, across time? Otherwise, sentences like "species on average were much more variable ($0.01 \leq CV \leq 0.03$), especially in Low rewards treatments where plants with pollinators and their rewards contributed large amounts of variability" provide little meaning.

Response: Thank you for noticing this error. We fixed this definition and provided a description of our various calculations at the beginning of the Results: **Stability** section on L248-258. We also provided a numerical benchmark for what we consider to be very stable considering that our ecosystems contained ~1000 units of biomass at the end of simulations. On L259 we state that “At the ecosystem level, all treatments were exceedingly stable ($CV < 0.001$).”

- The hypothesis the authors present is very difficult to follow, L250: "The control represents the hypothesis that the effects of pollinators on our systems are solely due to the total productivity of resources at the base of the food web that emerged due to pollination services during the original simulations." This is an important analysis, yet the control scenario appears to be presented as the actual working hypothesis rather than the simpler null hypothesis. "Productivity of resources" is not defined, and it is not described how this productivity emerges from "pollination services."

Response: We appreciate this point and now describe the purpose of our controls without distracting readers by referring to a hypothesis and more simply stating (L302-305) that, "This allows us to test whether the additional biomass produced by plants with pollinators is the sole cause of diversity, stability, and function in our multiplex networks or whether plant-pollinator feedbacks are required for these effects (Methods, S3)." Also, rewards productivity is now more clearly defined in **Table 1**. More detailed results in the style of Figs. 5-6 can be found in the supplementary information (**Fig. S7-8**).

- "On average, community species persistence..." (L253)---Average of what? What does "community species persistence" represent---this is a term only used once in the text?

Response: As described above and in the original manuscript, persistence is the fraction of species surviving to simulation's end. "Community species persistence" referred the fraction of species in the entire community that persist to the end of simulations, averaged across all simulations in the treatments. However, we realize this can be confusing because "community" often variously refers to different groups of species such as herbivores or pollinators. Therefore, we now refer to "ecosystem persistence" as the fraction of all initial species within a network that survives to simulation's end which is more clearly labelled in **Fig. 5** and described further in **Table 1**.

- The phrase "minor exception" is used frequently (L171; L178; L237; L258; L283)---I would advise against using the subjective qualifier, "minor," as "exception" on its own is adequate. (Otherwise, the authors give the impression that they are favouring a particular outcome.)

Response: We appreciate and agree with this important criticism. All instances of "minor" as a qualification have been removed (e.g. **L208, L331, L337**).

- "Former pollinators decreased in abundance despite access to up to twice the biomass previously available as rewards. In contrast, original herbivores increased in abundance in response to the increased vegetative biomass in the controls" (L264). This is an interesting and curious result that needs to be discussed further. Is it a direct consequence/artefact of modelling decisions or is it a surprising emergent property?

Response: We very much agree with this observation and suggestion. This result appears to be a surprising emergent property due to transient dynamics after mutualistic feedbacks have been turned off. However, the dynamical behavior is quite idiosyncratic between initial conditions such that we (so far) only feel confident reporting the observation and not in more specifically ascribing causality in the main text. However, we now discuss such causality in

our supplementary S3 on “Feedback and other controls.” To better understand the result, we will need a more targeted experimental design, which is outside of our current scope but is definitely of future interest.

- "More broadly, negative effects of mutualism consistently extend to none of our six other measures of ecosystem structure and function (diversity, persistence, abundance, productivity, consumption, guild-level stability) while positive effects of mutualism apply to all of our High rewards treatments and most of our Low rewards treatments" (L304). This sentence is very unclear---are the authors simply trying to say that, according to simulations, mutualism has a positive effect on the six measures?

Response: We agree this sentence is unclear. We were indeed trying to say that “mutualism has a positive effect on the six measures” but with a few exceptions. We address this problem following the previous suggestion by summarizing these overall results in **Fig. 4** and by revising the sentence (**L348-350**) for clarity using new terminology introduced in the revised manuscript: “Mutualism tends to increase stability and ecosystem function according to *all* of our measures in treatments with stronger mutualistic interactions and by several metrics in treatments with weaker mutualisms (Fig. 4).”

- New terms continue to appear in the discussion section, e.g., "omnivorous pollinators" (L331). Although I will add that the discussion is very nice from L358 onwards.

Response: We rewrote the beginning of the Discussion to conform with terminology presented earlier in the manuscript, especially in **Table 1**. Thank you for the compliment in regards to the rest of the Discussion. It has only been slightly revised, but, we hope, is now more impactful given clarified context earlier in the manuscript.

(4) Justify some of the seemingly arbitrary modelling decisions.

The methods section is generally well-written and comprehensive. There are, however, a few choices that should be justified to the reader.

- Why were 102 food webs used (L129; i.e., why not a round 100 or 120)?

Response: We aimed for 100 food webs but our procedure generated 102 and we felt little need to reduce our sample size though we now realize our choice should be better justified. We now clarify this on **L432-435**.

- Why were there twice as many animal-pollinator species than plant-with-pollinator species (L417)?

Response: There are twice as many pollinator species and plant species with pollinators because this is a widely observed central tendency in empirical networks, following our “trophic grouping” procedure described in **S1**. Furthermore, we explain on **L467-468** in Methods: **Network architecture** that we used a constant ratio “to maintain pollinators’ average resource availability in networks of increasing diversity (also see S1).”

- Why does connectance decrease with species richness with RO (L433) but is kept constant with RP (L438)?

Response: This is mostly an outcome of the decreasing connectance with increasing richness in pollination networks combined with the differences in RO and RP treatments. Connectance decreases from the original niche-model food web connectance of 0.1 with increasing species richness because added pollinators are only allowed to consume (link to) floral rewards in RO networks. This allows the decreasing connectance with increasing richness in pollination networks to reduce the connectance of the RO multiplex networks as they get larger. However, the added links between pollinators and non-floral resources in the RP treatment enable connectance to maintain a constant value with increasing richness. We addressed this in detail in **Fig. S3** and now expand our explanation in Methods: **Network architecture** on L479-497.

- How do the authors confirm that simulations are at a dynamical steady state (L535)? Does steady state include the possibility that some of the initial species have gone extinct?

Response: We extensively address this point and agree it is an important point to clarify. Some initial species virtually always go extinct during the simulations. Most go extinct relatively early in the simulations and few go extinct late such that we more formally consider simulations to achieve an approximate steady state. **Fig. 3** shows this. Following the reviewer's comment, we have clarified this further by adding an explanation to both **Table 1** and Methods: **Simulations**, where we additionally refer readers to **Fig. S2** which depicts robustness of all main text results to simulation length. **Table 1** specifies the definition of steady state as follows: "Formally, dynamics in which all species have constant abundance ($dB_i/dt = dR_i/dt = 0$ for all i). At the end of 5000 timesteps, our systems *approximate* steady-state dynamics (Fig. 3) as quantified by very small variability in total ecosystem biomass over the last 1000 timesteps of the simulations ($CVs < 0.0001$)."

- "One guild is just the rewards of all plants with pollinators" (L546)---Is this a guild of nectar?

Response: Yes, but it is only relevant for our production rate, consumption rate, and variability metrics. We specified this in **Table 1** and modified the Outputs section of Methods to clarify this on L603-604 and L616-617 "We calculated these metrics for the whole ecosystem (Fig. 4) and for seven guilds of species (Figs. 5-6)... When relevant (e.g. in Fig. 6), we considered the rewards biomass of all plants with pollinators as an eighth guild."

Minor comments

1. L111. The authors actually use the stochastic niche model, so this version should be cited rather than or in addition to the original niche model.

Response: The original niche model is a stochastic model that uses an algorithm employing two random numbers to set the properties of virtually every species. Several other niche models of food webs derived from the original are also stochastic. There are

several spatial stochastic niche models in the literature that are largely irrelevant for our purposes. We used the original niche model to generate the niche model food webs (**Fig. 2a**) and a novel method to link pollinators into the food web (**Fig. 2c**). Also see Methods: Network architecture.

2. L126. Why do the authors write "potentially antagonistic interactions"? Herbivory and carnivory are unambiguously antagonistic in that one individual is being negatively affected by the actions of another individual.

Response: We thank the reviewer for highlighting this seemingly subtle point, as it helped us clarify in the text how our work advances upon past theory. We do not agree that herbivory and carnivory are unambiguously antagonistic and cite these as examples of the logical inconsistencies that can arise when trying to classify species interactions in an effects-based framework. Specifically, the “overcompensation” literature describes how herbivory can increase the fitness in terms of biomass and reproductive output of individual plants, while the literature on “hydra effects” describes how mortality inflicted on prey can counterintuitively increase their population abundance despite predation being unambiguously detrimental to an individual prey. We cite these two examples (though there are many more available) to illustrate several advantages of our focus on modeling interaction mechanisms from which effects emerge instead of the effects themselves as has been frequently done in previous theory. We introduce these ideas in more depth on **L72-80** as “One significant problem is that classifying interactions based on positive and/or negative effects ignores logical inconsistencies such as when “antagonistic” herbivory or predation respectively increase plant fitness²⁶ or prey abundance²⁷ and when “mutualistic” pollinators parasitize plants by robbing their floral rewards without transferring pollen²⁴. We resolve such conflicts by modelling mechanisms by which organisms interact and allowing effects to emerge from the interactions rather than asserting such effects *a priori*. We do this by developing consumer-resource theory that has long been applied to food web theory^{16,28} and more recently applied to mutualistic interactions^{29,30} with success predicting pollinators’ foraging preferences in the field³¹.” We continue this theme throughout our Introduction (**L53-110**), and use it as a broad scheme for describing how we synthesize pollination and feeding interactions into a common consumer-resource framework.

3. L185. Or just "compensated"?

Response: Yes, that is more appropriate.

Reviewers' Comments:

Reviewer #1:

Remarks to the Author:

The authors have done a very good job revising the manuscript which has greatly improved in clarity. By introducing a mechanistic approach to modeling mutualistic and feeding interactions in multiplex ecological networks, this work notably advances the discipline of community ecology and reveal the effect of pollination mutualisms on the stability and function of food webs. The novel aspect is the synthesis of both types of consumer-resource interactions to better understand the controversial effects of mutualism on ecosystems at the species, guild, and whole-community levels.

I thus think the ms is now acceptable for publication in Nature Communications.

Reviewer #2:

Remarks to the Author:

The additional work that the authors have put into trying to answer my previous concerns is unquestionable, and I do appreciate their effort. I am convinced that there is merit in what the authors present in this manuscript, and both the results and the structure of the manuscript have been clearly improved relative to its previous version. Indeed, I was very pleased to see the additional controls presented in the ``feedback controls' section as well as the additional figures.

My only concern is in regards to how difficult it is to disentangle the effects of the different components of the analyses, which I had already expressed in my previous review. The authors have described the different components of the analyses much more clearly, but the use of random networks (or networks showing extreme cases of modularity and nestedness) would have been an informative addition to the manuscript that would have certainly helped me break down the different parts of the authors' conceptual framework. Moreover, I am admittedly a bit uneasy with the fact that I don't really know whether or not the empirically-informed network structure has any influence on the authors' results, or whether or not the results are mainly due to the design of the model. That said, I do understand that this is a significant amount of additional work, and that the manuscript is already very dense in terms of the number of reported results. All my other concerns have been answered satisfactorily.

Reviewer #3:

Remarks to the Author:

Review of revised "Pollinators in food webs: Mutualistic interactions increase diversity, stability, and function in multiplex networks" by Hale et al.

I am Reviewer 3 from the first round of reviews. I have read the revised manuscript and the authors' response letter. I appreciate the authors' thoughtful response to my comments and find that their edits and additions have considerably improved the manuscript. I have no further comments. I am supportive of publication in Nature Communications.

I sign my review: Phillip P.A. Staniczenko

REVIEWERS' COMMENTS:

Reviewer #1 (Remarks to the Author):

The authors have done a very good job revising the manuscript which has greatly improved in clarity. By introducing a mechanistic approach to modeling mutualistic and feeding interactions in multiplex ecological networks, this work notably advances the discipline of community ecology and reveal the effect of pollination mutualisms on the stability and function of food webs. The novel aspect is the synthesis of both types of consumer-resource interactions to better understand the controversial effects of mutualism on ecosystems at the species, guild, and whole-community levels.

I thus think the ms is now acceptable for publication in Nature Communications.

Reviewer #2 (Remarks to the Author):

The additional work that the authors have put into trying to answer my previous concerns is unquestionable, and I do appreciate their effort. I am convinced that there is merit in what the authors present in this manuscript, and both the results and the structure of the manuscript have been clearly improved relative to its previous version. Indeed, I was very pleased to see the additional controls presented in the ``feedback controls' section as well as the additional figures.

My only concern is in regards to how difficult it is to disentangle the effects of the different components of the analyses, which I had already expressed in my previous review. The authors have described the different components of the analyses much more clearly, but the use of random networks (or networks showing extreme cases of modularity and nestedness) would have been an informative addition to the manuscript that would have certainly helped me break down the different parts of the authors' conceptual framework. Moreover, I am admittedly a bit uneasy with the fact that I don't really know whether or not the empirically-informed network structure has any influence on the authors' results, or whether or not the results are mainly due to the design of the model. That said, I do understand that this is a significant amount of additional work, and that the manuscript is already very dense in terms of the number of reported results. All my other concerns have been answered satisfactorily.

Reviewer #3 (Remarks to the Author):

Review of revised "Pollinators in food webs: Mutualistic interactions increase diversity, stability, and function in multiplex networks" by Hale et al.

I am Reviewer 3 from the first round of reviews. I have read the revised manuscript and the authors' response letter. I appreciate the authors' thoughtful response to my comments and find that their edits and additions have considerably improved the manuscript. I have no further comments. I am supportive of publication in Nature Communications.

I sign my review: Phillip P.A. Staniczenko